# Multi-omic analyses reveal the unique properties of chia (*Salvia hispanica*) seed metabolism

Gerardo Alejo-Jacuinde [1], Héctor-Rogelio Nájera-González[1,4], Ricardo A. Chávez Montes [1,4], Cristian D. Gutierrez Reyes[2], Alfonso Carlos Barragán-Rosillo[1], Benjamin Perez Sanchez [1], Yehia Mechref [2], Damar López-Arredondo[1], Lenin Yong-Villalobos [1✉] & Luis Herrera-Estrella [1,3✉]

Chia (*Salvia hispanica*) is an emerging crop considered a functional food containing important substances with multiple potential applications. However, the molecular basis of some relevant chia traits, such as seed mucilage and polyphenol content, remains to be discovered. This study generates an improved chromosome-level reference of the chia genome, resolving some highly repetitive regions, describing methylation patterns, and refining genome annotation. Transcriptomic analysis shows that seeds exhibit a unique expression pattern compared to other organs and tissues. Thus, a metabolic and proteomic approach is implemented to study seed composition and seed-produced mucilage. The chia genome exhibits a significant expansion in mucilage synthesis genes (compared to *Arabidopsis*), and gene network analysis reveals potential regulators controlling seed mucilage production. Rosmarinic acid, a compound with enormous therapeutic potential, was classified as the most abundant polyphenol in seeds, and candidate genes for its complex pathway are described. Overall, this study provides important insights into the molecular basis for the unique characteristics of chia seeds.

[1] Department of Plant and Soil Science, Institute of Genomics for Crop Abiotic Stress Tolerance (IGCAST), Texas Tech University, Lubbock, TX 79409, USA. [2] Department of Chemistry and Biochemistry, Texas Tech University, Lubbock, TX 79409, USA. [3] Unidad de Genómica Avanzada/Langebio, Centro de Investigación y de Estudios Avanzados del Instituto Politécnico Nacional, Irapuato, Gto. 36821, Mexico. [4] These authors contributed equally: Héctor-Rogelio Nájera-González, Ricardo A. Chávez Montes. ✉email: lenin.yong@ttu.edu; luis.herrera-estrella@ttu.edu

The genus *Salvia*, which includes approximately 900 species, is the largest genus within the mint family (Lamiaceae)[1]. *Salvia* represents one of the most diverse genera of flowering plants comprising several species of economic importance. In general, they are cultivated worldwide for ornamental purposes, as culinary herbs, or as flavoring agents[2], and some are known for their therapeutic properties as antitumoral, anti-allergenic, antioxidant, or antimicrobial agents[3]. Furthermore, the consumption of seeds of one *Salvia* species commonly known as chia, *Salvia hispanica*, has gained popularity as a functional food or "superfood" in the last decades[4]. In addition to their high nutritional value, chia seeds contain several bioactive compounds with health-promoting properties[5]. Chia seeds are characterized by their significant amount of essential fatty acids, high protein content, antioxidant compounds, and high dietary fiber content[4,6]. Therefore, seeds of this species have great potential in the food industry to enhance the nutritional value of different products. Additionally, their derivatives (oil, flour, extracts, gum, etc.) can be used as foam stabilizers and emulsifiers or as a source of natural antioxidants to decrease lipid and protein oxidation in processed food[7]. In the current climate change scenario, *S. hispanica* has also been proposed as an alternative crop for its ability to grow in arid and semiarid environments[8].

*Salvia hispanica* seeds also represent a source of mucilage, a gelatinous sheath surrounding the seed when hydrated, to which several nutraceutical properties have been assigned. Seed mucilage is a matrix composed of polysaccharides, and chia seed mucilage mainly consists of xylose, glucose, arabinose, galactose, glucuronic acid, and galacturonic acid residues[9]. Chia mucilage is already used in the food industry as an additive in different preparations[10]. Because of its physical and chemical properties, some studies foresee a high potential of chia mucilage as a biopolymer in the cosmetic and pharmaceutical industries[11]. Additionally, chia seeds have high antioxidant activity comparable to commercial antioxidant products[12], suggesting that chia seeds represent an important dietary source of antioxidants. Different studies have reported the presence of compounds with antioxidant activity in chia seeds, including several flavonols and phenolic compounds[12–14]. Among antioxidants, rosmarinic acid (RA) was previously reported as the most abundant phenolic compound in chia seeds[14], which is already used in the food industry as a natural antioxidant or preservative. Thus, several efforts have been made to establish RA large-scale production using in vitro plant cultures[15].

Due to the nutraceutical properties and industrial applications of chia seed, there is an increasing interest in studying *S. hispanica* to understand the genetic basis of economically important traits. The genetic basis of seed-related features, such as mucilage production or polyphenols metabolism (e.g., RA biosynthesis) still needs to be discovered. Therefore, developing a high-quality reference genome is essential to characterize the molecular mechanisms behind the functional properties of chia seeds. Consequently, we produced a high-quality chromosome-level *S. hispanica* genome assembly in the present study using Oxford Nanopore and Hi-C sequencing. This improved assembly allowed the identification of most of the centromeric and some telomeric regions and the relocation of a chromosomal fragment compared to a previously reported chia genome assembly[16]. Our high-quality genome annotation revealed an expansion in the copy number of genes involved in seed mucilage biosynthesis compared to the genes previously reported in *Arabidopsis thaliana*. This study also identified potential regulators controlling mucilage production in chia seeds. Furthermore, we performed metabolic and proteomic analyses to describe seed-specific components. A combined approach integrating metabolomic pathways and transcriptomic data allowed the identification of

chia genes putatively involved in RA synthesis, one of the food industry's most used and economically important antioxidants.

## Results

**Assembly and pseudo-chromosome reference of *S. hispanica* genome.** A high-quality reference of the *S. hispanica* genome was obtained using a combination of Oxford Nanopore Technologies (ONT) ultra-long reads (46.68 Gb), Illumina paired-end reads (54.87 Gb), and Hi-C sequencing (25.13 Gb) for a total coverage of around 350× (Supplementary Table S1). The genome size of this *S. hispanica* Mexican variety (Supplementary Fig. S1) was estimated at 358.3 Mb by k-mer frequency analysis (Supplementary Fig. S2). Several assembly pipelines were evaluated to construct a high-quality reference chia genome (Supplementary Table S2). Among them, the NECAT assembler[17] produced the most contiguous (contig N50 = 5.37 Mb) and complete genome assembly (BUSCO complete 97.8%), which was chosen for further analyses. An improved assembly was obtained after six polishing steps using Illumina reads with Pilon[18] and subsequently processed with Purge Haplotigs to identify primary contigs[19]. Finally, Hi-C data was used to achieve a chromosome-level assembly using Juicer[20] and 3D-DNA[21] pipelines (Supplementary Fig. S3). This reference genome assembly was composed of 269 contigs integrated into 154 scaffolds with a total length of 351.98 Mb and a scaffold N50 length of 61.89 Mb (Fig. 1a; Supplementary Table S3), with 98.5% of the embryophyta dataset of BUSCO[22] determined as complete (Fig. 1b; Supplementary Table S4). Most of the assembly was anchored in six pseudo-chromosomes (345.52 out of 351.98 Mb; Supplementary Table S5). Analysis of the *S. hispanica* genome with Tandem Repeats Finder[23] and nhmmer (hmmer.org) allowed the identification of centromeres in five of the six chromosomes (Fig. 1a; Supplementary Table S6). The most common centromeric repeat in the chia chromosomes has a length of 171 bp, but variants of 167 and 174 bp were also identified (Supplementary Table S7). The typical plant telomeric repeat[24] was identified on one end of two chromosomes (TTTAGGG in chr1 and chr5; Supplementary Fig. S4). Other telomeric variants were placed on one end in two additional chromosomes (CTAAACC in chr2 and AAACCCT in chr6; Supplementary Fig. S4).

Alignment of our *S. hispanica* genome assembly with a previously published genome of an Australian chia variety[16] showed high collinearity between chromosomes of the two assemblies. The only significant difference between the two chia genome assemblies is that one end of the chromosome chr1 (a fragment of 4.6 Mb) of the Mexican variety was assigned to chromosome chr2 in the Australian variety (Supplementary Fig. S5), suggesting that this fragment was misassembled. This fragment was separated from the chromosome, and analyses using two different Hi-C datasets were performed to resolve the correct genomic location of this region. These analyses showed significantly higher Hi-C paired alignments between this fragment with chr1 than contacts with chr2, indicating that this genomic fragment effectively belongs to chr1 (Supplementary Fig. S6). A genome sequence of another *S. hispanica* Mexican cultivar (white seeds) was recently published[39]. This 4.6 Mb genomic fragment was also assigned to chr1 in this other *S. hispanica* genome, corroborating the result of our analyses. Genome alignment using this other Mexican cultivar also showed high collinearity between assemblies. Nevertheless, the results of this analysis indicate higher nucleotide identity between the assembly generated in this study with the Australian variety than the other Mexican variety published by Li et al.[39] (Supplementary Fig. S5).

**Genome annotation and methylation patterns in *S. hispanica*.** A total of 179.88 Mb was annotated as repetitive sequences, which

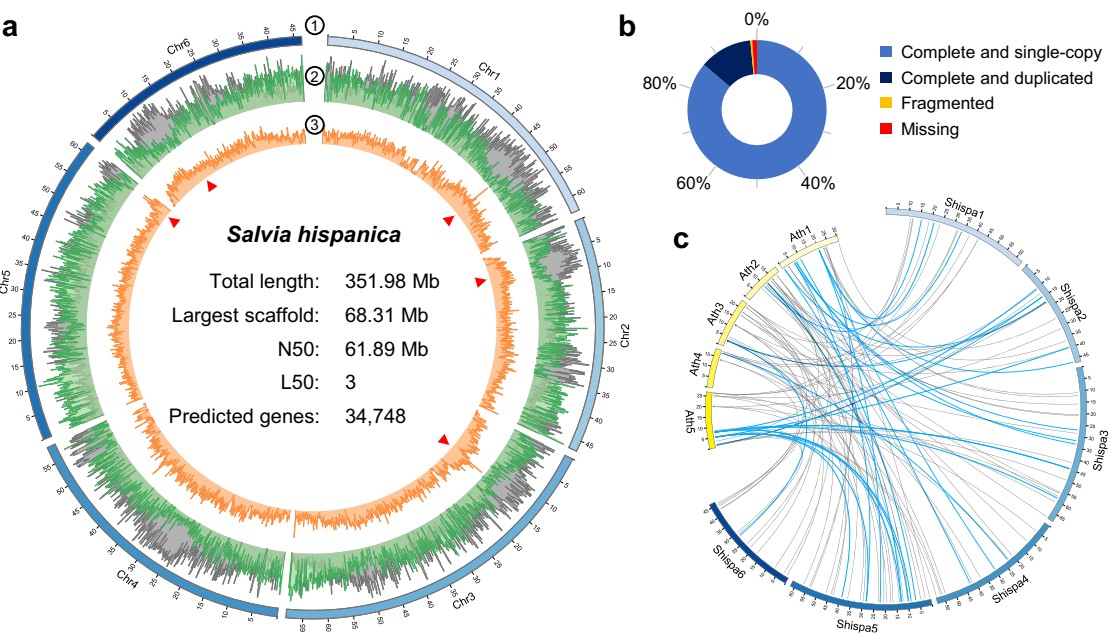

**Fig. 1 Chromosome-level reference of *S. hispanica* genome. a** Features and metrics of chia genome assembly. Track 1 depicts chromosomes number and their length (Mb), track 2 shows gene density (green) and repetitive elements (gray), track 3 methylation patterns (5mC distribution) per 100 kb windows. Approximate locations of centromeres indicated by red triangles, and final genome reference metrics (center). **b** High-level genome completeness indicated by conserved embryophyte orthologs. **c** Synteny with lines representing homologs genes for mucilage production between *A. thaliana* (Ath) and *S. hispanica* (Shispa). Blue lines highlight the expansion of mucilage synthesis genes in chia genome.

comprise around 51.1% of the *S. hispanica* genome. Most repetitive elements were classified as long terminal repeats (LTRs), occupying approximately 58.82 Mb of the genome. These are mainly composed of retroelements of the Gypsy and Copia classes (32.10 and 26.44 Mb, respectively). Repetitive sequences in the *S. hispanica* genome were masked before structural annotation. Gene models were predicted using MAKER-P pipeline[25,26] including ab initio and homology-based gene prediction methods. Annotation resulted in 34,748 protein-coding genes in the *S. hispanica* genome, with an average gene length of 2,828.35 bp and an average of 5.6 exons per gene. The set of predicted genes includes 97% of the BUSCO[22] embryophyta core as complete genes indicating a high level of genome completeness (Supplementary Fig. S7). This set of genes also has an average annotation edit distance (AED) of 0.15, indicating high agreement between supporting evidence and annotation.

To gain insights into the chia genome's methylation patterns, we determined cytosine methylation at carbon position 5 (5-methylcytosine, 5mC) using ONT. Raw ONT reads were processed according to the DeepSignal[27] pipeline. Approximately 15 million 5mC were obtained from four independent Nanopore sequencing runs, of which 92% were filtered as high-quality 5mC positions. The relative fraction of 5mCs identified in each sequence context for these high-quality positions was CG = 45%, CHG = 19.6%, and CHH = 35.4%, where H is any of the bases A, T, or C (Supplementary Fig. S8). Whole-genome analysis of 5mC reveals high DNA methylation levels in genomic regions with low gene density and high content of repetitive sequences while showing relatively lower methylation levels within gene-rich regions (Fig. 1a). In most cases, centromere positions were identified close to the maximum value of 5mC counts (Supplementary Fig. S9). DNA methylation levels (%) for chia genes were evaluated and showed that low levels were observed at the boundaries of the gene bodies near translation start and stop

codons in all sequence contexts, these levels increase in the gene body predominantly in the CG context and to a lesser extent in CHG and CHH methylation. In contrast, DNA methylation of repetitive DNA sequences such as transposable elements (TEs) showed higher DNA methylation levels in all three methylation contexts (Supplementary Fig. S8).

**Transcriptomic and metabolic profiles of chia seed.** Gene expression analysis of previously published RNA-seq data of key vegetative and reproductive growth stages of *S. hispanica*[28] showed a significantly distinct expression pattern for chia seeds (Fig. 2a); 20.75% of the chia genes have their maximum expression in seeds (CPM values Z score normalized) (Supplementary Fig. S10). Enrichment analysis of genes highly expressed in seed included a wide variety of biological processes, including seed-related categories like embryo development and abiotic stress terms such as response to salt stress and heat acclimation, among others (Supplementary Fig. S11).

To associate the transcriptional profiles with seed composition, we performed metabolomic and proteomic analyses of mature seeds. Lipidomic analysis of seeds identified almost 1000 lipid molecules that could be grouped into ten categories (Supplementary Fig. S12). The most abundant lipid class comprised triacylglycerols, of which the C18 acyl side chains with zero to eight unsaturated bonds were the most common (Supplementary Fig. S12). Metabolomic analysis of *S. hispanica* seeds identified a total of 1,984 features, of which the most abundant compound was (R)-(+)-rosmarinic acid, followed by oleamide, α-trehalose, 3-hydroxycoumarin, threonic acid, stachyose, buchananine, L-α-glyceryl phosphorylcholine, α-linolenic acid, L-phenylalanine, and palmitoleic acid (Supplementary Table S9). Metabolome results of chia seed analysis are summarized in Fig. 2b by the main functional group of each compound.

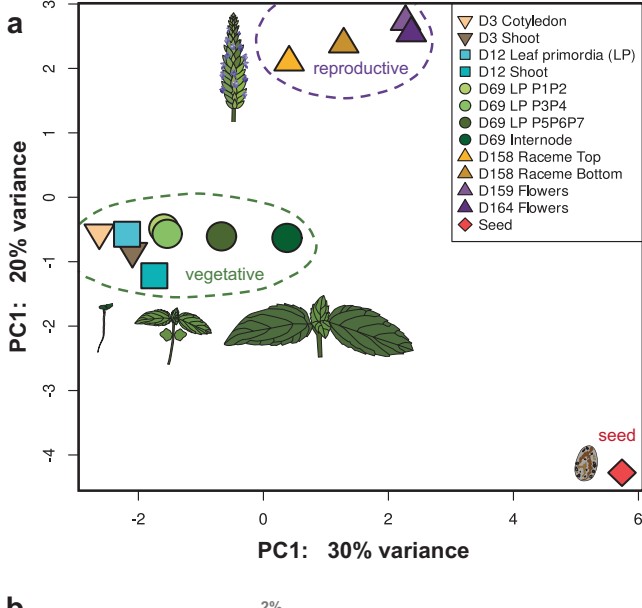

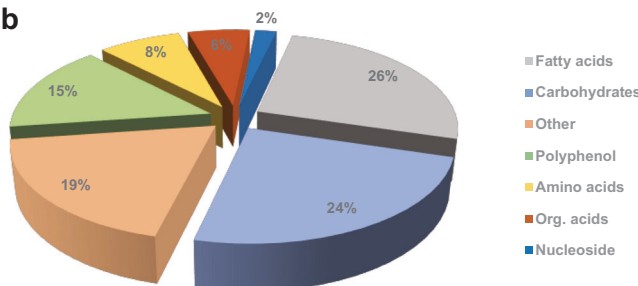

**Fig. 2 Chia seeds exhibit unique characteristics at gene expression and metabolic level. a** Principal component analysis of expression profiles across different tissues and stages during *S. hispanica* life cycle. Expression values were normalized and transformed to *Z* score prior to principal component analysis. **b** Metabolome analysis of dry chia seeds, chemical composition distributed in main functional groups. Cotyledon and shoot at 3 days (3D); leaf primordia (LP) and shoot at 12D; 1st and 2nd leaves (P1P2), 3rd and 4th leaves (P3P4), 5th, 6th and 7th leaves (P5P6P7), and internode at 69D; top or bottom half raceme inflorescence at 158D; flowers at 159D; flowers at 164D; dry seeds. Detailed description of the plant material in Supplementary Table S8.

**Rosmarinic acid biosynthesis pathway.** Polyphenols were mature seeds' fourth most abundant chemical group, with 15% of the chromatographic area. Among polyphenols, RA's abundance was significantly higher than other polyphenols (Fig. 3). The database MetaCyc was used as a reference to identify the metabolic pathway and enzymes involved in RA biosynthesis. Since the synthesis of RA in plants is a complex pathway involving several enzymes, substrates, and reaction products[29], we implemented a simplified nomenclature of its reactions, as indicated in Fig. 3. Briefly, it starts from L-tyrosine (denoted as reactions A1-2) or from L-phenylalanine (reactions B1-3), the product of these two pathways is converted into 4-coumaroyl-4'-hydroxyphenyllactate (reaction C1), and this precursor is hydroxylated in different positions to form RA. Still, this last conversion has not been entirely described (reactions D1-2 or E1-2). Almost all RA precursors and intermediates (9 out of 10) were detected in our metabolomic analysis (Fig. 3). Annotated genes in the *S. hispanica* genome were screened to identify genes with enzymatic function using the Ensemble Enzyme Prediction Pipeline (E2P2) software[30]. Chia genes encoding enzymes potentially involved in the RA pathway were identified by their EC number annotation. This screening identified 19 and 22 *S. hispanica* genes with putative enzymatic activity for the reactions A1-2, and B1-3, respectively (Fig. 4a; Supplementary Table S10). However, this analysis did not identify specific genes for the C1 step; then, candidate genes for this reaction (E.C. 2.3.1.140) were searched using previously reported *Salvia miltiorrhiza* and *Melissa officinalis* rosmarinic acid synthase genes[31,32] using BLASTP. This screening identified 57 candidate genes for reaction C1. Fifty-six candidate genes were identified for the last reaction of RA synthesis, reactions D1-2/E1-2 (Fig. 4a; Supplementary Table S10). To better define candidates for the last two RA synthesis steps, gene expression was examined across different tissues and developmental stages (indicated as 'life cycle' samples in Supplementary Table S8). This analysis showed that 7 out of these 113 genes had their maximum expression level in seed compared to other tissues (Fig. 4b). Therefore, these results allowed the identification of 48 genes as primary candidates for RA synthesis in *S. hispanica* seeds (19, 22, 5 and 2 genes for reactions A1-2, B1-3, C1, and D1-2/E1-2, respectively; Supplementary Table S10), suggesting that the RA pathways in encoded by several multigene families.

**Seed mucilage-related genes and their regulation.** Genes putatively involved in *S. hispanica* seed mucilage production and release were identified using a set of mucilage-specific genes previously reported for *Arabidopsis thaliana* seeds[33]. A homology-based BLASTP approach identified 108 mucilage-related genes in *S. hispanica* (Supplementary Table S11 and Fig. S13). Mucilage-related genes with a significant number of copies in the chia genome include *CSLA2* (6 copies), *CESA2* (5 copies), and *ADK1*, *COBL2*, *MUM4/RHM2*, and *RRT1* (all of them with four copies) (Supplementary Fig. S13). These genes were classified by their proposed function in a) mucilage synthesis, b) mucilage modification, c) mucilage stabilization, and d) mucilage secretion, or involved in e) cell wall synthesis and modification, f) hormone synthesis and perception, and g) transcriptional pathways (Fig. 5a). This analysis showed that genes that belong to the mucilage synthesis category were significantly expanded in the chia genome (28 genes compared to 12 reported in *A. thaliana*). Of these, only two pairs of genes were identified as tandem gene duplications, the homologs of the *Arabidopsis* mucilage synthesis genes *CSLA2* and *GAUT11* (Supplementary Table S11). Using available RNA-seq data, gene expression of mucilage-related genes was examined during seed development at 3, 7, 14, 21 and 28 days after flowering (DAF)[34]. This analysis showed that 72 out of 108 mucilage-related genes exhibit maximum expression at 7 DAF (Fig. 5b). All mucilage gene categories were expanded in the chia genome regarding the number of genes in *Arabidopsis* except for the transcription factors (TFs) group. Therefore, a gene regulatory network analysis was conducted to identify TFs that could regulate mucilage production in chia seeds. A weighted gene correlation network was constructed using the RegEnrich[35] pipeline, which then was filtered to retain the top 5% of the highest TF – gene correlations that included mucilage-related genes (Fig. 6). The resulting network comprises 16 potential regulatory genes of chia seed mucilage production, including members of the AP2, ARF, bHLH, bZIP, C3H, GRF, NAC, MIKC MADS, MYB, and TALE families of transcription factors. Notably, this analysis identified homologs of five TFs previously reported as regulators of mucilage metabolism in *Arabidopsis* seed (Fig. 6).

Proteins associated with seed mucilage represent essential components for its structure. Therefore, a proteomic analysis was carried out to identify proteins in *S. hispanica* seed mucilage.

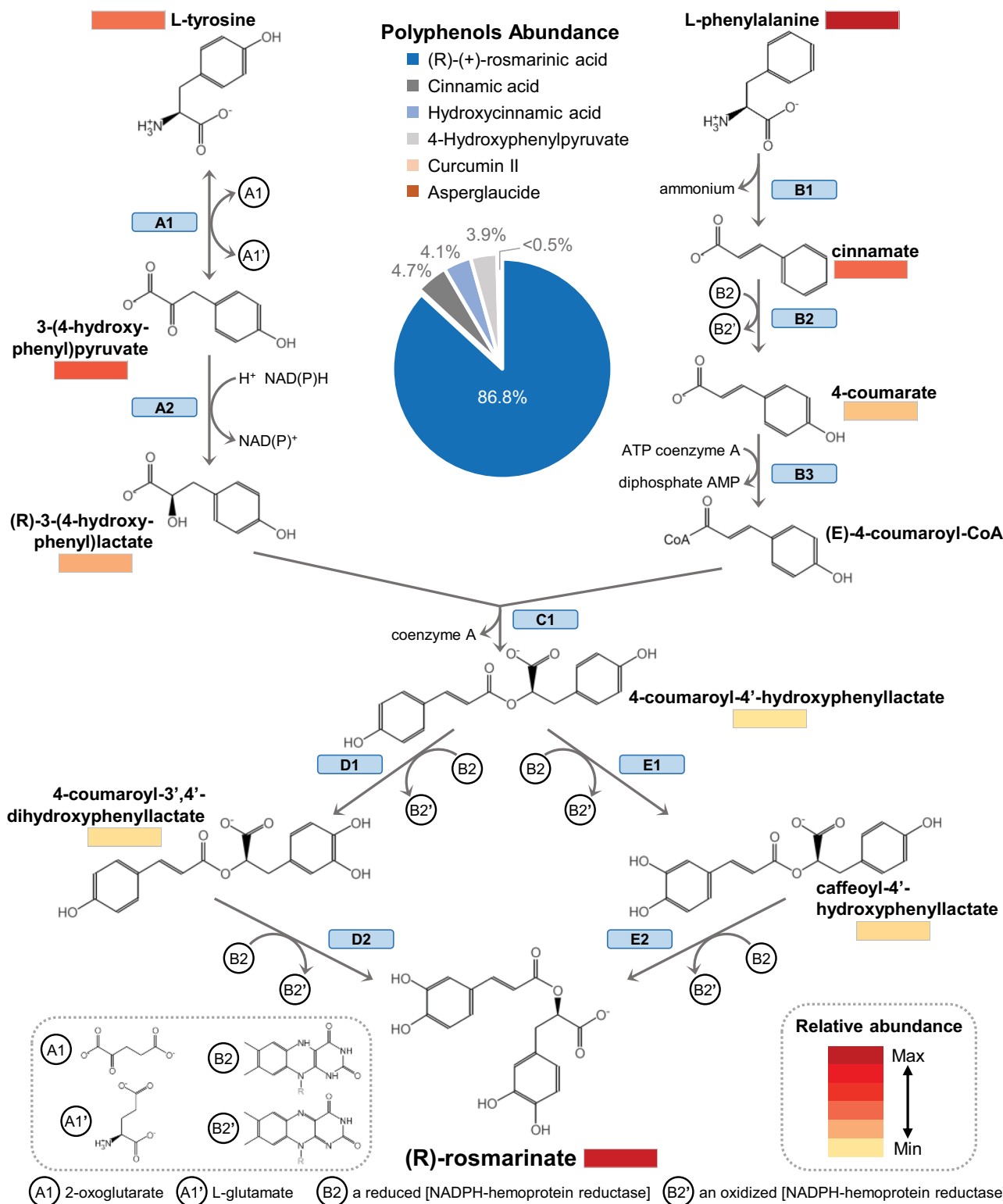

**Fig. 3 Rosmarinic acid and its precursors are significantly accumulated in chia seeds.** Metabolome analysis indicated rosmarinic acid (RA) as the most abundant compound in chia seeds. RA is also accumulated in much higher amount compared to other polyphenols (pie chart; center). The biosynthesis pathway of RA is represented, indicating the precursors identified by the metabolic analysis and their relative abundance in chia seeds. Nomenclature to indicate pathway reactions is depicted as blue boxes. Numerical source data for polyphenols is provided in Supplementary Table S9.

Mucilage was extracted from imbibed chia seeds, digested with trypsin, and analyzed by liquid chromatography coupled to mass spectrometry (LC-MS). Analysis of the peptides identified by LC-MS allowed the identification of 95 proteins in the chia mucilage sample. These proteins were functionally annotated using

PANZER, InterPro, and *A. thaliana* homolog description to generate a classification according to their protein family and biological activity (Supplementary Data S1). We also identified 11 glycoproteins using Byonic software which also determined the glycans attached to the proteins (Supplementary Table S12).

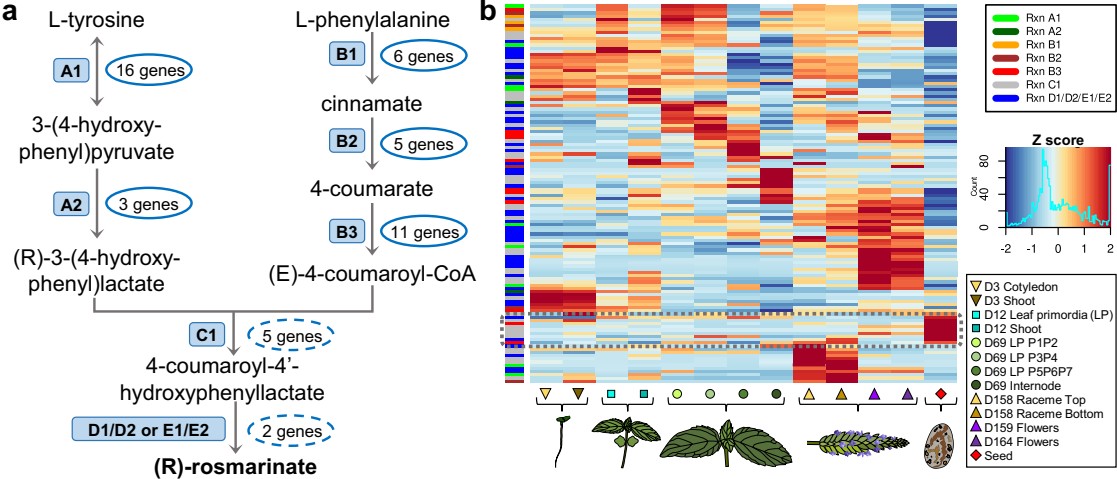

**Fig. 4 Candidate genes regulating the synthesis of rosmarinic acid in chia seeds. a** Simplified representation of rosmarinic acid (RA) synthesis pathway and the number of genes in *S. hispanica* with potential enzymatic activity for each reaction. **b** Approach to reduce the number of candidates genes for the last part of the RA pathway selecting the genes highly expressed in chia seeds (dashed rectangle). Expression profiles (*Z* score normalized) of RA genes across different tissues during the life cycle of *S. hispanica*. Dashed ellipses indicated primary candidates for reactions C1, D1/D2, and E1/E2. Sample descriptions and abbreviations are similar to those in Fig. 2.

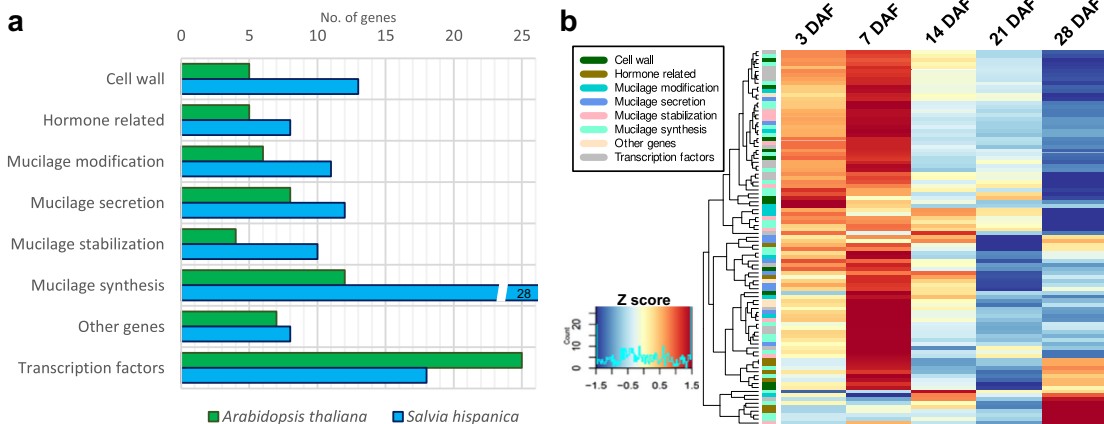

**Fig. 5 Seed mucilage-related genes were significantly expanded in the chia genome. a** Number of mucilage-related genes in *Arabidopsis thaliana* (green) and their homologs identified in *S. hispanica* (blue). Genes were grouped according to their proposed function or type of gene. **b** Expression profile of mucilage-related genes during seed development in *S. hispanica*. Gene expression (*Z* score normalized) at different days after flowering (DAF).

Glycoproteins identified in chia mucilage participate in different functions, including at least 3 of the so-called dirigent (DIR) proteins involved in secondary metabolism. Expression analysis indicates that genes coding for proteins classified into carbohydrate and lipid metabolism are mainly expressed 7 DAF, correlating with the maximum expression of the previously identified mucilage-related genes (Supplementary Fig. S14), whereas mucilage proteins related to abiotic stress, like LEA are mainly expressed at the end of the chia seed development. Finally, to remove possible contaminants in the chia mucilage proteome, proteins with a relative abundance <0.1% were filtered out (Supplementary Table S13). This analysis resulted in a high-confidence set of 37 proteins in the chia seed mucilage, represented mainly by seed storage proteins, proteins related to lipid metabolism, late embryogenesis abundant (LEA) proteins, and some related to secondary metabolism (Table 1).

## Discussion
Recently, there has been an increased interest in developing chia as a main crop for human consumption and as a source of bioactive compounds. Worldwide production of chia seeds has increased in the last decades[36], and some studies predict that the global chia market will continue growing in the following years[37]. However, information and resources for its genetic improvement still need to be improved. This study provides critical genomic resources for understanding the genetic basis of relevant seed-related traits in *S. hispanica*. A high-quality *S. hispanica* genome reference was generated using ONT and Illumina sequencing and chromosome-level scaffolding with Hi-C data. The final genome reference shows high contiguity, with more than half of the assembly in only three scaffolds (pseudo-chromosomes) and an N50 metric of 61.89 Mb (Supplementary Table S3). Compared with previously reported *S. hispanica* genomes sequenced using PacBio[16], the assembly generated in this study has a significant improvement in capturing repeat-rich regions, such as centromeres (Fig. 1a; Supplementary Fig. S9), and the identification of telomeric repeats for some chromosomes (Supplementary Fig. S4). Highly repetitive regions are difficult to reconstruct, and genome assembly using PacBio sequencing could fail in large repetitive regions[38]. The successful reconstruction and identification of centromeric regions in our chia genome indicate that using ONT ultra-long reads allowed us

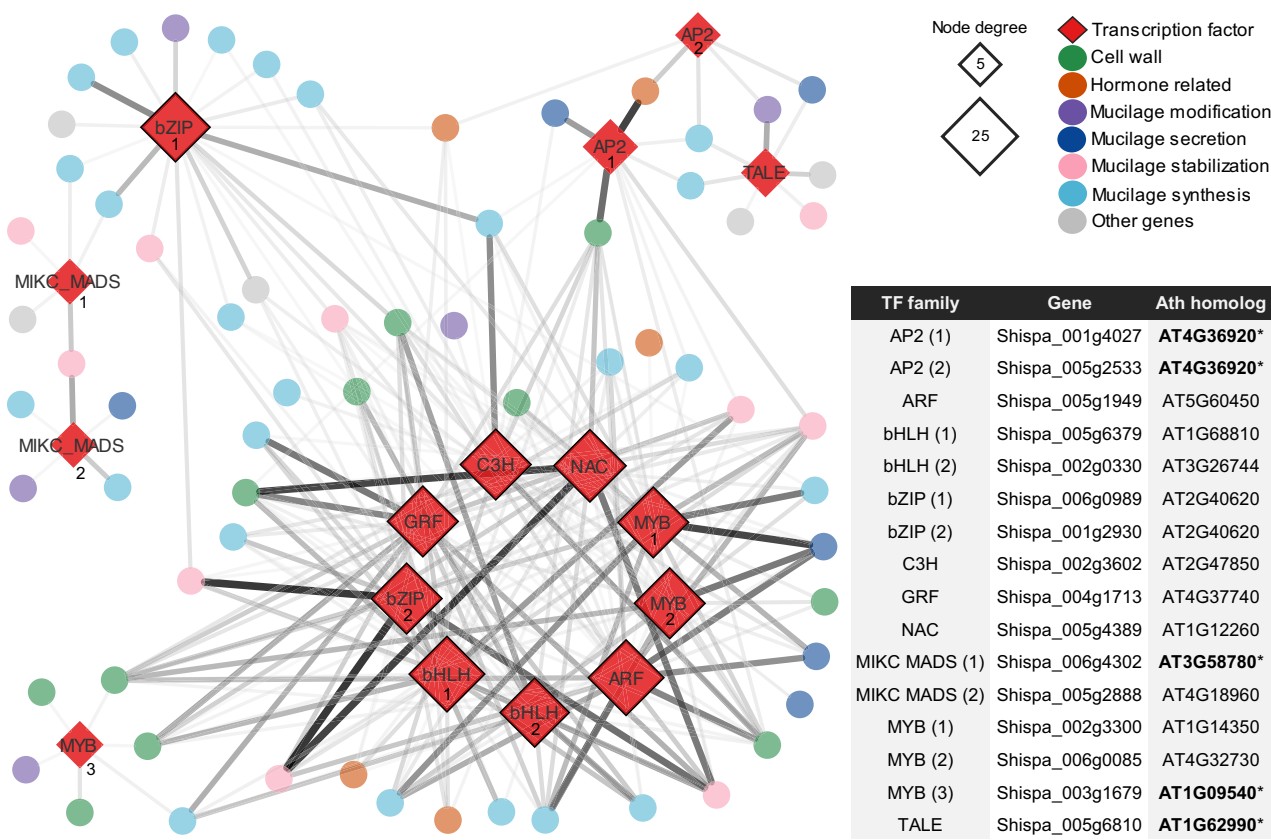

| TF family | Gene | Ath homolog |
|---|---|---|
| AP2 (1) | Shispa_001g4027 | **AT4G36920*** |
| AP2 (2) | Shispa_005g2533 | **AT4G36920*** |
| ARF | Shispa_005g1949 | AT5G60450 |
| bHLH (1) | Shispa_005g6379 | AT1G68810 |
| bHLH (2) | Shispa_002g0330 | AT3G26744 |
| bZIP (1) | Shispa_006g0989 | AT2G40620 |
| bZIP (2) | Shispa_001g2930 | AT2G40620 |
| C3H | Shispa_002g3602 | AT2G47850 |
| GRF | Shispa_004g1713 | AT4G37740 |
| NAC | Shispa_005g4389 | AT1G12260 |
| MIKC MADS (1) | Shispa_006g4302 | **AT3G58780*** |
| MIKC MADS (2) | Shispa_005g2888 | AT4G18960 |
| MYB (1) | Shispa_002g3300 | AT1G14350 |
| MYB (2) | Shispa_006g0085 | AT4G32730 |
| MYB (3) | Shispa_003g1679 | **AT1G09540*** |
| TALE | Shispa_005g6810 | **AT1G62990*** |

**Fig. 6 Transcription factors proposed as regulators of mucilage production in chia seeds.** Gene regulatory network analysis allowed the identification of transcription factors (TFs) with significant correlations with mucilage-related genes. Proposed TFs acting as regulators for mucilage production in *S. hispanica* are indicated in red diamonds. The table indicates the *Arabidopsis* homolog of each TF, and those shown in bold (*) have been previously reported as involved in seed mucilage synthesis. Network inference was conducted using the RegEnrich package. Genes are represented as circles (color indicates their proposed function), and edge weight means the significance of the correlation (a thick line depicts a strong TF-gene correlation). Genes showed as MICK MADS are also annotated as SHP in some studies.

**Table 1 Summary of the type of proteins identified in chia seed mucilage.**

| Group | Number of proteins |
|---|---|
| Seed storage prot | 12 |
| Lipid metabolism | 8 |
| LEA | 7 |
| Secondary metabolism/Neolignans | 2 |
| Seed maturation proteins | 2 |
| Carbohydrate metabolism | 1 |
| Detoxification enzyme | 1 |
| Proteolysis | 1 |
| Lectin | 1 |
| Cell wall | 1 |
| Not defined | 1 |
| **TOTAL** | **37** |

to cover genomic regions with a high proportion of repeats. For example, the over 600 repeats present in the centromeric region of chr5 are poorly represented in the same chromosome of the previously reported chia PacBio assembly. Interestingly, the centromere of chr5 exhibits a large tandem arrangement where most of the centromeric repeats are next to each other, separated by only two bases.

An improved annotation of the *S. hispanica* genome was obtained in the present study with the prediction of 34,748 protein-coding genes (compared to 31,069 genes reported in the

Australian variety[16]) and an almost complete set of conserved embryophyte orthologs (97% complete BUSCOs compared to 93.2% of the PacBio assembly; Supplementary Fig. S7). Comparison of the genomes of the Mexican chia variety with that of the Australian variety[16] indicated high agreement in some characteristics of the *S. hispanica* genome. Both studies determined a GC content of around 36.6%, a similar proportion of repetitive sequences with the LTR elements Gypsy and Copia as the most abundant. However, the estimated genome size was slightly larger for the Mexican variety (358.3 Mb) than the Australian one (354.5 Mb). Genome alignment indicated high collinearity between Mexican and Australian chia genomes (Supplementary Fig. S5); however, a 4.6 Mb fragment was located in different chromosomes in these two assemblies (Supplementary Fig. S5). Our analyses demonstrated that this genomic fragment is part of chr1 (Supplementary Fig. S6). Furthermore, a recently published chia genome, which is also at chromosome-level assembly, assigned this same genomic fragment to chr1 supporting our results[39].

DNA methylation is an epigenetic modification in the cytosine base important in regulating gene expression and genome stability in plants, influencing plant developmental processes and responses to environmental factors[40,41]. Single base resolution methylomes of various plant species have been reported[42–44], providing information on the dynamics of DNA methylation pathways[42,44], molecular function, and genomic distribution[40,45]. *Salvia hispanica* methylation patterns show global DNA percentage proportions like those previously reported for other plant

methylomes (Supplementary Fig. S8), for which the most predominant DNA methylation context is CG, followed by CHH and CHG[42–44]. The analysis of DNA methylation distribution along gene bodies showed a similar behavior as that reported for angiosperms[43,46], where low DNA methylation levels are detected around the start/stop translation sites. Specifically, DNA methylation showed a significant increase in the methylation density in TEs, concordant with the role of DNA methylation in silencing such repetitive elements[47,48]. Further analysis of DNA methylation patterns may provide insights into how DNA methylation regulates chia developmental changes, gene expression, and responses to abiotic and biotic stresses. Moreover, global methylation studies may be essential to understand secondary metabolism in chia, as previously reported for *S. miltiorrhiza*, where chemical inhibition of DNA methylation using 5-azacytidine significantly enhanced phenolic acid accumulation by altering DNA methylation patterns in gene promoters, including the promoter of the RA synthase gene[49].

*Salvia hispanica* is mainly valuable by the nutritional value and health-promoting properties of its seeds[50]. A previous study highlighted the unique distinct seed transcriptome compared to other developmental stages in *S. hispanica*[28]. Indeed, chia seeds have a unique expression pattern compared to other tissues or growth stages (Fig. 2a). Chia seed analyzed in the present study confirms that fatty acids (FA) are the most abundant component (26% of the total chromatographic area; Fig. 2b). Lipid content in chia seeds can range from 20% to 34%, and it is particularly rich in polyunsaturated FA, mainly α-linolenic acid[6]. Our lipidomic analysis showed that α-linolenic acid is one of the most abundant seed compounds, at least 3.5 times higher than linoleic acid (Supplementary Table S9). Other authors reported concentrations of α-linolenic acid from 59.9 to 63.2% and of linoleic acid from 18.9 to 20.1% for chia seeds indicating a 3:1 ratio (ω-3:ω-6)[51]. Because of its high content of ω-3 fatty acids, there have been several studies to identify and elucidate the genetic pathway involved in chia lipid biosynthesis[16,34,39,52].

Species of the *Salvia* genus are characterized as rich sources of polyphenols with multiple therapeutic applications[53]. Metabolome analysis of mature chia seeds determined that 15% of the total chromatographic area corresponds to polyphenolic compounds (Fig. 2b). The most abundant compound identified in *S. hispanica* seeds was rosmarinic acid (RA), also the most abundant caffeic acid dimer reported in *Salvia* species[53] (Supplementary Table S9). A previous report determined that RA has concentrations of up to 0.92 mg/g in chia seeds[14]. The principal activities of RA include antioxidant, anti-inflammatory, astringent, antimutagen, antibacterial and antiviral properties[54]. The high content of RA in chia seeds could suggest a defense role against pathogens and prevent lipid peroxidation due to its antioxidant activity. The synthesis of RA in plants is complex, with a non-linear pathway involving at least eight different reactions[29]. A detailed metabolic analysis identified all precursors and intermediates of the RA pathway in chia seeds except for 4-coumaroyl-CoA (Fig. 3). Besides, we conducted an exhaustive gene search to identify candidate genes for every reaction of this pathway (Supplementary Table S10). Since the last part of the RA synthesis has not been completely elucidated, many enzymes potentially performing the final hydroxylation steps were identified in *S. hispanica* genome. Thus, genes with putative enzymatic function for these reactions were examined for higher expression in seeds compared to other tissues. Overall, we identified and proposed a set of 48 genes for the synthesis of RA in *S. hispanica* seeds (Fig. 4). Knowledge of the genes participating in the RA pathway has an important potential for future bioengineering to increase the production of RA or its derivatives in emerging crops, such as chia.

Seed mucilage is another valuable component of *S. hispanica* with many potential applications. It is used in the food industry as an ingredient replacer in baked, dairy, and meat products[55]. Chia mucilage exhibits exceptional physical properties and can be suitable for preparing nanocomposites or the controlled release of drugs[56]. Moreover, its potential applications also include generating edible films as substitutes for synthetic packaging[57]. However, despite its industrial importance and possible future technological applications, the detailed composition and the genetic components involved in chia mucilage synthesis have yet to be described. This study identified genes putatively involved in *S. hispanica* seed mucilage metabolism and their expression during seed development. Current knowledge about genes participating in seed mucilage synthesis, stabilization, secretion and modification, and how they are regulated have been described in the model plant *Arabidopsis thaliana*[58,59]. Results of the homology analysis indicated that mucilage-related genes were expanded in the *S. hispanica* genome, mainly in the mucilage synthesis category (Fig. 5a). This gene group was significantly expanded in the chia genome compared to *Arabidopsis* (Fig. 1c). Mucilage-related genes with a significant number of copies in the chia genome exhibit diverse functions, including maintenance the correct structure of the adherent mucilage layer (*CSLA2*), contributing to mucilage synthesis (*CESA2*), participating in cellulose deposition into secondary cell wall structures (*COBL2*), and genes required to synthesize cell wall pectin polysaccharides (*MUM4* and *RRT1*)[60–64]. Mucilage has been described as a specialized secondary cell wall[65]. Current knowledge indicates that mucilage accumulates in the outermost epidermal layer of the seed coat, called mucilage secretory cells (MSC)[59]. In *Arabidopsis*, these MSC exhibit a significant production of cell wall material at seven days post anthesis (embryo torpedo stage)[66]. Interestingly, most mucilage-related genes identified in the *S. hispanica* genome have their maximum expression at early seed development, precisely at seven DAF (Fig. 5b). Furthermore, several of the most abundant mucilage-associated proteins, which could represent essential components for mucilage structure, also exhibit maximum expression during early development (Supplementary Fig. S14).

Since several *Arabidopsis* TFs involved in seed mucilage production did not have a direct homolog in *S. hispanica* genome (Supplementary Fig. S13), a regulatory network analysis was carried out to identify possible transcriptional regulators of chia seed mucilage. This analysis identified 16 TFs as important regulatory nodes for mucilage-related genes in *S. hispanica* (Fig. 6). The *Arabidopsis* homologs of five of these 16 putative regulators were previously described as TFs regulating seed mucilage production (including *AP2*, *SHP1*, *MYB61*, and *KNAT7* types)[33,59]. Additional candidate regulators include TFs of the ARF, bHLH, bZIP, C3H, GRF, NAC, MIKC MADS, and MYB families. Members of these TFs families have been reported as essential components of the transcriptional pathways controlling seed coat differentiation, mucilage synthesis, and modification[58]. Most of these regulators have their maximum expression at seven DAF (Supplementary Fig. S15), correlating with the highest expression of mucilage-related genes in *S. hispanica* seeds (Fig. 5b). Visualization of gene expression data of their *Arabidopsis* homologs (https://bar.utoronto.ca/eplant/) indicated that most of the proposed regulators are mainly expressed between early heart to torpedo stages during embryo development. A preliminary characterization of *S. hispanica* embryo development showed a high correlation between the early developmental stages of chia and *Arabidopsis* (Supplementary Fig. S16). Together, all these

findings are consistent with current knowledge about seed mucilage production in the model plant *Arabidopsis*. Understanding the genetic basis for mucilage production in *S. hispanica* could be used to bioengineer mucilage with improved characteristics for industrial or technological purposes.

Although proteins are present in a much lower proportion than polysaccharides in mucilage, proteins have an essential role in the dynamics and functionality of cell walls[67]. The present study identified 37 proteins in the *S. hispanica* seed mucilage (Table 1; Supplementary Table S13), 12 classified as seed storage proteins comprised 67% of the protein abundance. Previous reports indicated that globulins correspond to the main protein fraction identified in chia seeds[68]; this study determined that they are also present in seed mucilage, represented mainly by 11 S globulins (Supplementary Data S1). The type of proteins identified in chia mucilage is like groups previously reported in *Arabidopsis* seed coat mucilage[69]. Compared to *Arabidopsis* seed mucilage, chia mucilage shows a higher number of seed storage and LEA proteins and a much lower number of carbohydrate and antioxidant enzymes. Chia mucilage proteins exhibiting outstanding activities include DIR proteins characterized for modulating cell wall metabolism during development and in response to abiotic and biotic stress conditions[70,71]. Specifically, DIR proteins are involved in lignan and lignin biosynthesis. Lignans have an essential role in plant defense responses against pathogens[72,73], whereas lignin is a crucial component of plant vascular systems and for maintaining cell wall turgor and porosity[74,75]. In plants, DIR proteins have been described to have a role in seed coat protective neolignan byosinthesis[71], abiotic[76,77], and biotic stresses[78,79]. Seed mucilage has been reported to influence seed germination and seedling establishment[80], and DIR proteins could contribute to these mucilage-proposed functions during biotic and biotic stresses.

Currently, *S. hispanica* is mainly cultivated for seed production, but its leaves represent an additional source of diverse metabolites with antioxidant and antimicrobial properties[81]. There have been some efforts to study relevant secondary metabolism pathways in chia leaves, such as terpenoid biosynthesis[82]. Undoubtedly, molecular information will contribute to studying chia's unique secondary metabolism and help establish this species as a significant crop worldwide. The genome, proteins, genes, and pathways described in this work could provide the basis for improving the production and the quality of chia compounds and derivatives.

## Methods

**Plant material, nuclei and HMW DNA isolation**. *Salvia hispanica* plants of a Mexican variety (Supplementary Fig. S1) were grown in a Conviron walk-in chamber programmed at 26 °C for 16 h of light the first 3 months, then changed to 12 h of light to induce flowering. Nuclei of *S. hispanica* were isolated as described by Peterson (1997)[83]. Briefly, young leaves from a single *S. hispanica* individual (~2 grams) were collected and ground in liquid nitrogen. Grounded tissue was then transferred to a new falcon tube into Liquid nitrogen and stored at -80 °C until use. Nuclei isolation was carried out by adding 40 ml of Nuclei Isolation Buffer (NIB) and incubated for 10 min at 4 °C. The lysate was filtered through 3 layers of Miracloth, collected in a tube on ice, and centrifuged for 10 min at 1000x *g* at 4 °C to precipitate nuclei. The precipitated nuclei pellet was then resuspended in Percoll 70% in NIB and centrifuged at 600x *g* for 10 min. The nuclei pellet was washed with NIB and centrifuged at 600x *g* twice. NIB excess was removed by pipetting, and the nuclei pellet was either used for high molecular weight (HMW) DNA isolation or flash frozen with liquid nitrogen and stored at -80 °C.

**Genome sequencing (ONT, Illumina, Hi-C)**. Following the manufacturer's instructions, an aliquot of 200 μl of nuclei pellet was used for HMW DNA isolation using the Nanobind Plant Nuclei Big DNA kit (Circulomics, USA). Genomic DNA integrity was assessed using Agilent 4200 TapeStation System (Agilent Genomics). The purified HMW DNA was prepared for ONT sequencing using the ligation sequencing kit SQK-LSK109 following manufacturer instructions (ONT, Oxford, UK). Libraries were loaded onto R9.4.1 flow cells and sequenced using a MinION Mk1C instrument (ONT, Oxford, UK). Illumina short-read sequencing was

conducted by Novogene (Beijing, China) on an Illumina NovaSeq 6000 in PE150 mode. For the Hi-C samples, flash frozen nuclei pellet (~20 μl) was thawed on ice and crosslinked following the protocol described in the Crosslinking–Low Input section of the Arima-HiC 2.0 Kit User Guide for Mammalian Cell Lines (Arima Genomics). Crosslinked nuclei suspension was then eluted in 130 μl of elution buffer and fragmented to an average fragment size of 550–600 bp using a Covaris E220 Focused Ultrasonicator. Then, fragmented DNA was size-selected to a size distribution >400 bp using magnetic beads (Magbind Total Pure NGS, Omega BIO-TEK) and used as input to prepare proximally ligated DNA as directed in the Arima-HiC 2.0 protocol manual. Hi-C scaffolding libraries were prepared using the Swift Accel NGS 2 S Plus Kit following the manufacturer's instructions.

**Sequencing data processing and genome assembly**. The Illumina reads were quality trimmed (for adaptors, low-quality bases) using TrimGalore (v0.6.6; https://github.com/FelixKrueger/TrimGalore). Jellyfish[84] (v2.3.0) was used for *k*-mer (21-mer) frequency distribution based on the quality trimmed Illumina reads. The output histogram was used to estimate genome size with the R package findGSE[85] (v1.94). Raw ONT signals (fast5 files) were base called using Guppy (v5.0.16), and sequencing adapters were trimmed off using Porechop (v0.2.4; https://github.com/rrwick/Porechop). The processed ONT reads were error corrected and assembled using NECAT[17] (v0.0.1) software. The output assembly was polished using six iterations of Pilon[18] (v1.24) with Illumina data. Primary contigs were identified using Purge Haplotigs[19] (v1.1.2). The contigs were scaffolded using Juicer[20] and 3D-DNA[21] pipelines with Hi-C data. The scaffolds were manually curated to create a genome assembly with chromosome-length scaffolds. At every step of this process, the quality of the assemblies was evaluated using BUSCO[22] (v5.4.3) with embryophyte and eudicots (odb10) datasets and Illumina data alignment rate using Bowtie2[86]. Centromeric tandem repeats were identified using Tandem Repeats Finder[23] (v4.09.1) and nhmmer (v3.3.2; hmmer.org). Before defining their locations, candidate centromeric regions were manually compared with methylation (5mC counts), gene density, and repetitive DNA content. Telomeric repeats were identified using Telomere Identification toolKit (v0.2.0; https://github.com/tolkit/telomeric-identifier) searching the typical plant telomeric sequence ('TTTAGGG').

**Chromosomal fragment relocation analysis**. Minimap2[87] (v2.24) was used to align the genome reference generated in this study with the previously reported chia genome[16]. The online tool D-Genies[88] was used to visualize whole-genome alignments. A fragment of around 4.6 Mb was identified as the main difference between these assemblies. Detailed analyses were carried out to determine the correct genomic location of this region. First, this region was split and considered as another contig to perform Hi-C data analysis using two independent datasets (Hi-C data generated in this study and an available data SRA Accession: DRX325318). Hi-C contact heatmaps were generated and visualized using Juicebox[20] (v1.11.08). Finally, the contact frequencies and number of Hi-C paired alignments determined this fragment's most probable genomic position.

**DNA methylation profiling**. Methylation calling using ONT data was performed using the DeepSignal-plant pipeline[89]. ONT reads were converted to single fast5 files by the multi_to_single_fast5 command included in the ont_fast5_api package. The nanopore reads and fast5 files were then preprocessed using the tombo preprocess and tombo resquiggle commands. Next, the deepsignal_plant call_mods command was used to perform the DNA methylation calling running the 5mC model trained using *Arabidopsis thaliana* and *Oryza sativa* R9.4 1D reads (model.dp2.CNN.arabnrice21_120m_R9.4plus_tem.bn13_sn16.both_bilstm.epoch6.ckpt). High quality 5mC positions from 4 different nanopore runs were obtained after the DeepSignal-plant pipeline, and only those 5mC positions present in at least 2 replicates were retained for further data analysis.

**Genome annotation**. Genome assembly was annotated through three rounds of MAKER-P[25,26] (v3.01.03). First round used 1) as EST evidence transcripts obtained by aligning a total of 66 available RNA-seq libraries[28,34,52,82] using HISAT2[90] (v2.2.1) and reconstructing the transcripts from the alignment file using StringTie[91] (v2.2.1); 2) as protein evidence the plants OrthoDB obd10 dataset available at the ProtHint's github webpage (https://github.com/gatech-genemark/ProtHint), filtered to keep only species belonging to the Lamiales order; 3) as the repeats file the de novo repeat sequences obtained using RepeatModeler (v2.0.3) with Repeat-Masker (http://www.repeatmasker.org; v4.1.2-p1), Dfam[92] (v3.6), and an LTR_retriever[93] (v2.9.0) run of LTR_Finder_parallel[94,95], and ltr_harvest[96] results. Second round used as input the maker gff3 file from the first round, a SNAP hmm file obtained from round 1 gene models, and the Augustus[97] gene models from a BUSCO[98] run following Daren Card's method (https://darencard.net/blog/2017-05-16-maker-genome-annotation/; Augustus section) with the eudicots_odb10 dataset. The third round was run as the second round, using round 2 output files as inputs. The round 3 gff3 file was used to recreate the CDS sequences using a custom Perl script, which were translated to protein sequences using the transeq program from the EMBOSS suite (v6.6.0). Genome annotation completeness was evaluated using BUSCO[22] with embryophyte and eudicots (odb10) datasets.

**Genome functional annotation**. Gene Ontology annotations were assigned using a simplified version of the MAIZE-gamer pipeline[99]. Briefly, annotations were assigned as the GO annotations of the BLASTP reciprocal best hits versus Araport11 and UniProt Swiss-Prot proteins from nine plant species (*Glycine max, Oryza sativa* subsp. japonica, *Populus trichocarpa, Solanum lycopersicum, Sorghum bicolor, Vitis vinifera, Brachypodium distachyon, Physcomitrium patens,* and *Chlamydomonas reinhardtii*), the GO annotations from an InterProScan[100] (v5.57-90.0) analysis, and the GO annotations from the PANNZER2[101] functional annotation webserver with a PPV value of at least 0.5. All these GO annotations were merged into a non-redundant gaf file.

**Seed metabolite and lipid extraction**. Chia seeds (1 g of dry seeds) were grounded in liquid nitrogen and freeze-dried overnight. Subsequently, 300 mg of chia seed powder was extracted with 3 ml of methanol:chloroform mixture (1:1 v/v), sonicated for 5 min and centrifuged at 4000 rpm for 10 min. Both supernatant and insoluble phases were collected to process further. An aliquot of 2 ml of supernatant was dried in a speed vacuum and later reconstituted in 200 μL of 95% ethanol HPLC grade. The insoluble phase was dried with nitrogen flow overnight and then reextracted with 1 ml water:methanol mix (60:40 v/v). This polar extract was filtered with 0.2 μm reconstituted cellulose filters and injected into Vanquish Flex UHPLC chromatography system coupled to mass spectrometry detector Exploris 240 (Thermo Scientific).

**Lipidomic analysis**. Lipid extract of chia seeds was analyzed with a $150 \times 2.1$ mm C18 column (Thermo Scientific) using a gradient method of water 0.1% formic acid (phase A) and isopropyl alcohol 2 mM ammonium formate (phase B). Chromatographic conditions were as follows: Started 10% B from 0 to 3 min, then increased B concentration to 50% from 3 to 7 min and further 50–100% B from 7 to 20 min. The 100% B was kept for 6 min, then equilibrated to 10% B from 26 to 30 min. The column temperature was 50 °C, and flow was kept constant at 200 μl/min. Detection was achieved with both positive and negative ionization modes at 4500 V and 1200 V, respectively. Data-dependent MS/MS (DDA) was acquired for the most intense 20 peaks in each full scan with dynamic exclusion applied for 10 successive scans. Data analysis was done using LipidSearch 5.0 software (Thermo Scientific) with LipidMaps online database[102].

**Metabolomic analysis**. Polar metabolite extract was injected into a reverse phase C18 column and separated using a gradient method of water 0.1% formic acid (A) and methanol 0.1% formic acid (B). The initial flow was set to 200 μl/min, 5% B for 4 min, then increased to 100% B in 13 min. This condition was maintained for 4 min and then set back to 5% B to re-equilibrate for 4 min. Independent runs were used for positive and negative detection modes with MS/MS DDA. Raw files were uploaded to Compound Discoverer 3.3 software (Thermo Scientific) and analyzed using metabolomics pipeline. Result tables from positive and negative ionization modes were joined and further investigated to remove duplicate identifications. Metabolite identification was made using online databases mzCloud, KEGG, and BioCyc and further curated manually.

**Proteomic analysis**. Mucilage extraction was performed according to Tsai et al.[69]. Briefly, seeds (2 g of dry seeds; 3 replicates) were soaked in water 1:10 w/ v and placed in an orbital shaker for 1 h. Then transferred into liquid nitrogen and freeze-dried for 48 h. Mucilage was separated from the seed with a small mesh and collected to quantify its protein content. For proteomics analysis, 300 mg of dried chia mucilage was mixed with 30 ml of 50 mM ammonium bicarbonate (ABC) buffer. Then, an aliquot of 100 μl of the resuspended mucilage was placed into a filter microtube with 100 kDa cut-off and incubated at 90 °C in a water bath for 15 min for protein denaturation. Protein digestion was completed on the surface of the filter mentioned above. Initially, dithiothreitol (DTT) was added to reduce the protein disulfide bonds by incubation at 60 °C for 45 min. Then, the cysteine residues of the denatured and reduced proteins were alkylated with iodoacetamide (IAA) at 37 °C for 60 min. The sample was digested with trypsin enzyme according to manufacturer instructions in a ratio of 1:25 w/w for 18 h at 37 °C. Finally, the tryptic digests were separated from the mucilage by centrifugation at 10,000 rpm for 10 min. The recovered tryptic digests were dried in a centrivap and desalted using C18 Top Tips. The samples were resuspended in 20 μl mobile phase (98% water, 2% acetonitrile, 0.1% formic acid). Samples were then analyzed into an Ultimate 3000 nanoLC with a PepMap RSLC C18 chromatography column (3 μm, 100 Å, 75 μm x 15 cm). Mobile phases used were A: water 0.1% formic acid (FA) and B: acetonitrile:water (80:20 v/v) 0.1% FA. The method started 2% B for up to 5 min and then increased to 25% B in 105 min, 25-40% from 105 to 125 min, 40–95% B from 125-126 min and back to 2% B at 128 min for 6 min. Detection was carried out in an Orbitrap Exploris 240 (Thermo Scientific, USA) in the positive mode set to 120,000 in the orbitrap resolution. DDA MS/MS was set to $5 \times 10^3$ intensity threshold, and charge states 2-8. Raw data files were uploaded and processed with Proteome Discoverer 2.5 and Byonic software. Protein identification was made using the predicted proteins in the *S. hispanica* genome, using as filters FDR ≤ 0.01 and at least 3 unique peptides in the sequence. We use the

Byonic software to confirm the identity of the glycoproteins. Byonic search for glycosylations along the protein peptide backbone, allowing an accurate identification of the glycans attached to the proteins.

**Gene expression analysis; genes for rosmarinic acid and mucilage biosynthesis**. Publicly available RNA-seq data during *S. hispanica* life cycle, including different tissues and growth stages[28] was retrieved from the EMBL-EBI ArrayExpress (experiment number E-MTAB-5515), and additional RNA-seq dataset of *S. hispanica* during seed development[34] was retrieved from the NCBI SRA database (accession number PRJNA196477). Samples during life cycle included: cotyledon and shoot at 3 days (3D); leaf primordia (LP) and shoot at 12D; 1st and 2nd leaves (P1P2), 3rd and 4th leaves (P3P4), 5th, 6th and 7th leaves (P5P6P7), and internode at 69D; top or bottom half raceme inflorescence at 158D; flowers at 159D; flowers at 164D; and dry seeds. Samples during seed development included: developing seeds at 3, 7, 14, 21, and 28 days after flowering. A detailed description of each tissue type and growth stages included in these transcriptomes is provided in Supplementary Table S8. RNA-seq raw sequences were quality trimmed (for adaptors, low-quality bases) using TrimGalore (v0.6.6). Gene expression analysis during chia life cycle was quantified using kallisto[103] (v0.44), normalized (CPM). The average CPM value between replicates was transformed to Z score prior to principal component analysis (plotMDS limma package[104]). The Ensemble Enzyme Prediction Pipeline was used to identify putative enzymes and classify them according to their predicted catalytic functions[105]. Candidate genes for RA biosynthesis were identified using previously reported Enzyme Commission numbers for its synthesis in plants (2.6.1.5, 1.1.1.237, 4.3.1.5, 1.14.13.11, 6.2.1.12, 2.3.1.140)[29]. Homologs of *S. miltiorrhiza* and *M. officinalis* rosmarinic acid synthase (GenBank: FJ906696.1 and FR670523.1, respectively) were used for the EC 2.3.1.140 reaction[31,32]. The information for converting 4-coumaroyl-4'-hydroxyphenyllactate to (R)-rosmarinate was taken from MetaCyc (EC 1.14.14.-). Expression levels of candidate RA genes during life cycle were quantified using kallisto[103] (v0.44) and normalized using DESeq2 package[106]. The mean expression across replicates was Z score normalized and used to analyze across different tissues and stages during life cycle. Previously reported genes for seed mucilage metabolism in *A. thaliana*[33,59] were used to identify homolog genes in the *S. hispanica* genome. Mucilage-related genes were identified using BLASTP with the set of *A. thaliana* genes as query. To avoid misidentification of mucilage-related genes, criteria to select homologs in *S. hispanica* included: bitscore ≥ 80, alignment of the total length of the query sequence ≥ 80%, and identity between query and subject sequences ≥ 50%. Candidate genes were manually classified by their best hit according to the proposed function in *A. thaliana*. Expression levels of mucilage-related genes were quantified using kallisto[103] (v0.44), and $\log_{10}$(TPM) values were Z score normalized to analyze expression during seed development. The webserver PlantTFDB (v5.0; http://planttfdb.gao-lab.org/) was used to identify TFs in *S. hispanica* genes[107]. The RegEnrich[35] (v1.4.0) package was used to construct gene regulatory networks with GRN (i.e., random forest algorithm) as the inference method and the list of predicted TFs as potential regulators. The output network object was ordered by edge weight to select important regulatory nodes and filtered to retain the top 5% with the highest correlations. The resulting network was manually analyzed to identify all gene interactions involving mucilage-related genes, and finally, candidate regulators were determined by high node degree.

**Statistics and reproducibility**. Leaves from a single *S. hispanica* individual were collected to perform the Nanopore, Illumina, and Hi-C sequencing. For the seed metabolic, lipidomic, and proteomic analyses three biological replicates were used. Statistical significance was calculated using software integrated methods. A *p* value of <0.05 was considered statistically significant for the gene ontology enrichment analyses.

**Reporting summary**. Further information on research design is available in the Nature Portfolio Reporting Summary linked to this article.

## Data availability

All data supporting the results of this study are included in the manuscript and its supplementary files. The whole-genome sequencing data and genome assembly are available at NCBI under BioProject PRJNA978134 and at CoGe with genome ID 66177. The gene models and genome annotation generated in this study can be accessed from www.depts.ttu.edu/igcast/Staff/Data_availability.php. The seed mucilage proteomic data was deposited MassIVE and ProteomeXchange under the identifiers MSV000092476 and PXD043875, respectively. Access details for the publicly available RNA-seq data used in the study are provided in the Methods section. All other relevant data are available upon request to the corresponding authors.

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

## Acknowledgements

This work was funded by the Governor University Research Initiative program (05-2018) from the State of Texas.

## Author contributions

G.A.-J., L.Y.-V. and L.H.-E. conceived and designed the study; G.A.-J. and L.Y.-V. generated biological material for genome sequencing; G.A.-J. performed the genome assembly; L.Y.-V. performed the methylation analysis; R.A.C.M. performed the genome annotation; H.-R.N.-G., C.D.G.R. and Y.S.M. performed the lipidomic, metabolic and proteomic analyses; G.A.-J. and R.A.C.M. conducted the study of the rosmarinic acid and mucilage biosynthesis; A.C.B.-R. and B.P.S. conducted chia seed development characterization; G.A.-J., L.Y.-V., R.A.C.M., H.-R.N.-G., D.L.-A., and L.H.-E. analyzed data; G.A.-J., L.Y.-V. and L.H.-E. wrote the manuscript. All authors read and approved the final manuscript.

## Competing interests

The authors declare no competing interests.
