## [Peer Review File · Communications Biology]

Reviewers' comments:

Reviewer #1 (Remarks to the Author):

Chia (*Salvia hispanica*) is an important niche crop with increasing economic, nutritional, health-related, and industrial applications. Structural and functional genomic analyses are essential and timely. This manuscript documents sequencing, assembly, annotation, and functional genomics of a reference genome for chia. The methodology and results appear to be appropriate and sound. It is not the first published whole-genome sequencing of the chia genome; the authors appropriately compare their genomic analysis to that of Wang et al. (2022). However, they failed to cite or compare their analysis to that of Li et al. (2023, <https://www.frontiersin.org/articles/10.3389/fpls.2023.1102715/full>), which was published 20 March 2023 but has been available online in pre-print form since late 2022. Their focus on gene expression in developing chia seeds, the biochemical composition of mature seeds as it relates to metabolic pathways and gene duplication, and rosmarinic acid and mucilage production are more extensive than in previous publications with novel discoveries, and are relevant given the economic importance of these aspects for chia.

The most serious deficiencies I found in this manuscript are the lack of comparison of the results reported here with those of Li et al. (2023), the lack of clarity regarding seed developmental stages for RNA-seq data and seed composition analysis, and the lack of information regarding statistical analysis of significant correlations. I highlighted where in the manuscript the authors need to address these deficiencies in my line-by-line comments.

Line-by-line comments:

Line 18: The following statement in the abstract is inaccurate: "the molecular basis of relevant chia traits remains to be discovered." As they discuss in the introduction section, authors of previous publications on chia (*Salvia hispanica*) have addressed the molecular basis of several traits. The authors cited one genomic study (Wang et al., 2022) that addressed the molecular basis of several nutritional and health-related traits, but they failed to cite another genomic study (Li et al., 2023, <https://www.frontiersin.org/articles/10.3389/fpls.2023.1102715/full>) that also addressed the molecular basis of such traits. The authors should correct and update the abstract to accurately reflect the results of previous research.

Line 21: The authors should delete the word "completely" and add the term "developing" to read as follows: "Transcriptomic analysis showed that developing seeds exhibit distinct expression patterns" Mature seeds are mostly dormant in gene expression. Gene expression in seeds takes place predominantly in developing or germinating seeds. Also, the patterns are not "completely" distinct in that some housekeeping genes undoubtedly are expressed in "other organs and tissues." The word "distinct" is sufficient to make the point.

Line 24: The term "expansion" is a relative term. Is the expansion in gene copy number relative to other species of the genus *Salvia*, or some other plant species or group of species? Later in the manuscript, the authors identified this expansion as relative to *Arabidopsis*, as indicated in parentheses in line 197, and in lines 202–204. If that is what they intend to convey in the abstract, they should make that comparison clear in the abstract.

Lines 70–71: My comment for line 24 also applies here.

Lines 101–110: The authors should expand this paragraph, or add another paragraph, to compare their research with the genome published by Li et al., 2023, which has been available online in pre-print form since late-2022, and was published in final form on 20 March 2023 (see

<https://www.frontiersin.org/articles/10.3389/fpls.2023.1102715/full>).

Lines 141–147: I presume this paragraph refers to gene expression in developing seeds, although this information is lacking. Presumably, this information could refer to developing seeds at various stages of post-fertilization seed development, mature seeds (which are mostly dormant for gene expression), or germinating seeds. Although Figure 2a provides some additional clarity, entirely in the image but not the caption, the reader should not have to try to decipher this information from Figure 2a. Instead, the authors need to clarify the developmental stage (or stages) of the seeds from which they obtained mRNAs in the narrative here, in the caption for Figure 2a, and in the Methods section. They also did not fully clarify to what extent their RNA-seq analysis is based on data from experiments they conducted and/or on data they obtained from previous publications, such as the research reported in their references 28–31. In referencing this previous RNA-seq research, the authors likewise did not clarify the developmental stage of the seeds analyzed in previous studies. I looked in the Methods section and could not find the information there. The fact that in lines 148–157 they refer to analysis of proteins and lipids from mature seeds further underscores the need for them to clarify the developmental stages of the seeds for RNA-seq analysis in lines 141–147.

Lines 159–160: Because these sentences begin a new section, the authors should clarify that the polyphenols to which they refer in this section are from mature seeds, not simply seeds (I presume this is the case, but I cannot be certain).

Lines 179–182: Here again the authors need to clarify the stage (or stages) of seed development. In line 180, they stated that “gene expression was examined across different tissues and stages.” However, in line 181, they state that “7 of these 113 genes had their maximum expression in seed” without clarifying which stage or stages of seed development they examined. They later, in lines 201–202 and 222–226, provided the sort of clarity regarding developmental stage that they need to provide here.

Lines 205–207, 344, and 468–469: The authors need to specify in statistical terms what they mean by “significant correlations.” Because this is a statistical term, they need to clearly state what they mean by “significant.” They need to describe the statistical test they used to derive this conclusion of significance and the probability threshold they chose for significance and why they chose this threshold. Figure 6 and its caption do not clarify the statistical nature of this correlation. In line 344, the authors used the relative phrase “more significant correlations” but failed to state the comparison. Do they mean “more significant correlations” relative to Arabidopsis or some other comparison, or do they simply mean “significant correlations”? The Methods section is also deficient in this respect. In lines 468–469, the authors simply state that “the output network was manually filtered to retain only significant correlations” without defining what a significant correlation is.

Lines 287–293: Here again the authors need to clarify the stage or stages of seed development to which they are referring in these sentences. I presume, but cannot be certain, that in lines 287–288 that are referring to developing seeds at some unidentified number of days post-fertilization, and in lines 289–293 to mature seeds, but I cannot be certain of these presumptions. They need to clarify the stages of the seeds throughout this paragraph.

Lines 298–299: Once again, the authors need to clarify that the seeds to which they are referring in this paragraph are mature seeds (if my presumption is correct).

Methods Section – The authors need to add a paragraph describing their RNA-seq analysis in developing seeds and the preparation of plant material for this analysis, and how they analyzed RNA-seq data derived from other publications.

Reviewer #2 (Remarks to the Author):

The passage discusses a study on chia (*Salvia hispanica*), an emerging crop considered a functional food with multiple potential applications. The study generated an improved chromosome-level reference of the chia genome and described its methylation patterns and refined genome annotation. Transcriptomic analysis showed that chia seeds have completely distinct expression patterns compared to other organs and tissues. The study implemented a metabolic and proteomic approach to study seed composition and seed-produced mucilage. The chia genome exhibits a significant expansion in mucilage synthesis genes, and gene network analysis revealed potential regulators controlling seed mucilage production. Rosmarinic acid, a compound with enormous therapeutic potential, was identified as the most abundant polyphenol in seeds, and candidate genes for its complex pathway were identified. Overall, the study provides important insights into the molecular basis for the unique characteristics of chia seeds.

I did not see any information in the Materials and Methods section of the article on how batch effects between different transcriptome datasets from different sources were handled.

It is not clear from the article how PCA analysis was performed on transcriptome data from different time points. Further information on the specific methods and procedures used would need to be provided to answer this question.

The discussion section is too complicated and some parts can be merged into the results. Please simplify the discussion section.

Seed mucilage synthesis related genes were predicted in genome of chia (*Salvia hispanica*), but only identified the expanded gene family compared to *Arabidopsis thaliana* rather than its relative species.

Line 92

Please point out the concrete assembly that anchored into chromosomes.

Line103-106

In addition to using Hi-C data to identify assembly errors, is it possible to supplement with ONT data to jointly determine assembly errors at this locus?

Line134-139

I cannot understand the meaning of this sentence.

Line 141: Insufficient evidence in gene expression analysis associated with seed development when identify the unique expression pattern. Why not implement the RNA-seq data based on more than one development stage because of these samples used in specific days after flowering (DAF).

Line144

Why use CPM as the measure of gene expression? Perhaps FPKM and TPM would be more appropriate.

Line148-149

What is the number of biological replicates for transcriptome, metabolome, and proteome? I did not find the corresponding quantity in the article.

Line162-164

Please provide relevant literature.

Line168-169

Can LC-MS be used to validate substances not found in the metabolome?

Line174-177

Please provide relevant literature.

Line196-197

What method was used to test the significance of this expansion?

Line 200

The expression of mucilage-related genes is important for the investigation of seed development in chia, but there were confused in the concrete sampling time (3DAF, 7DAF and 14DAF etc.)

Reviewer #3 (Remarks to the Author):

This study generated an improved chromosome-level reference of the chia genome and several transcriptomic, metabolic and proteomic data with the highlight of seed-specific components. These data are useful and essential to conduct more studies related to dissecting genetic basis of diverse traits in chia. However, I have some main concerns for the improvement of this manuscript.

1) As mentioned in introduction section, seed mucilage is a matrix composed of polysaccharides, surrounding the seed. if yes, mucilage is biosynthesized or accumulated in seed coat? it seems that transcriptomic, metabolic and proteomic data from the seed coat tissue. personally, I think that these data are critical and direct evidence to understand the molecular or genetic basis that why Chia seed produces mucilage.

2) As authors claimed, the levels of genome DNA methylation (including CG, CHG and CHH) are so high in gene body regions compared to intergene regions and TE regions. This seems unusual.

3) Is there any certain link between mucilage and protein identified from seed (shown in Table 1), These proteins are involved in mucilage biosynthesis, or they usually mix with mucilage?

4) In discussion section, author should highlight how improved genome data from previous one, only resolve the repeat regions? how about the 34,748 protein-coding genes, same to previous identification?

Point-by-point response reviewers

REVIEWER 1:

We thank this reviewer for the comments to clarify and improve our manuscript. We agreed with the reviewer who kindly indicated us missing information in some statements of our manuscript, and also to encourage us to prepare and provide a complete description with all the details of the methodology performed in the present study.

Comment Line 18: *The following statement in the abstract is inaccurate: “the molecular basis of relevant chia traits remains to be discovered.” As they discuss in the introduction section, authors of previous publications on chia (*Salvia hispanica*) have addressed the molecular basis of several traits. The authors cited one genomic study (Wang et al., 2022) that addressed the molecular basis of several nutritional and health-related traits, but they failed to cite another genomic study (Li et al., 2023, <https://www.frontiersin.org/articles/10.3389/fpls.2023.1102715/full>) that also addressed the molecular basis of such traits. The authors should correct and update the abstract to accurately reflect the results of previous research.*

Response: We thank the reviewer for this correction. We agree that other authors have previously described the molecular basis of some chia traits. However, most available chia molecular studies have focused on seed lipid metabolism. In this work, we studied the molecular basis of other commercially important chia traits, seed mucilage, and rosmarinic acid biosynthesis. Therefore, in this revised version of our manuscript, our statement was modified to include the approach of our study: “However, the molecular basis of some relevant chia traits such as seed mucilage and polyphenols remains to be discovered.” Additionally, we now cite the paper of Li et al. (2023) among the references of studies elucidating chia genetic pathways.

Comment Line 21: *The authors should delete the word “completely” and add the term “developing” to read as follows: “Transcriptomic analysis showed that developing seeds exhibit distinct expression patterns” Mature seeds are mostly dormant in gene expression. Gene expression in seeds takes place predominantly in developing or germinating seeds. Also, the patterns are not “completely” distinct in that some housekeeping genes undoubtedly are expressed in “other organs and tissues.” The word “distinct” is sufficient to make the point.*

Response: We thank the reviewer for this comment. Our results show that seeds have a distinct and unique expression pattern compared to other tissues and growth stages of the chia life cycle. As the reviewer mentioned, some genes share or show a similar expression in other organs or tissues. However, a large percentage of genes are seed-specific or exhibit their maximum expression in seeds, as shown in **Supplementary Fig S9**, which clearly shows that chia seed, like many other plant seeds, presents a unique expression profile particularly related to the accumulation of storage compounds, accumulation of seed-specific secondary metabolites and the acquisition of desiccation tolerance that are absent in most if not all other plant tissues. To be more precise, the statement in the Abstract was corrected to “Transcriptomic analysis showed that seeds exhibit a unique expression pattern compared to other organs and tissues”.

Comment Line 24: The term “expansion” is a relative term. Is the expansion in gene copy number relative to other species of the genus *Salvia*, or some other plant species or group of species? Later in the manuscript, the authors identified this expansion as relative to *Arabidopsis*, as indicated in parentheses in line 197, and in lines 202–204. If that is what they intend to convey in the abstract, they should make that comparison clear in the abstract.

Response: We agree with the reviewer that expansion is a relative term, and there was missing information in when we used the term “expansion” in the abstract. Therefore, in our manuscript's revised version, we corrected the Abstract: “The chia genome exhibits a significant expansion in mucilage synthesis genes (compared to *Arabidopsis*), and gene network analysis revealed potential regulators controlling seed mucilage production.”

Comment Lines 70–71: My comment for line 24 also applies here.

Response: We thank the reviewer for this observation. A similar response to the last comment. In this revised version of our manuscript, we added that information in the last paragraph of the Introduction: “The high-quality genome annotation revealed an expansion in the copy number of genes involved in mucilage biosynthesis compared to the genes previously reported in *Arabidopsis thaliana*. This study also identified potential regulators controlling mucilage-related genes in chia seeds.”

Comment Lines 101–110: The authors should expand this paragraph, or add another paragraph, to compare their research with the genome published by Li *et al.*, 2023, which has been available online in pre-print form since late-2022, and was published in final form on 20 March 2023 (see <https://www.frontiersin.org/articles/10.3389/fpls.2023.1102715/full>).

Response: We would like to thank the reviewer for this suggestion. As we explained in the letter to Editor David Favero, although Li *et al.* published a pre-print of a chia genome last year, the manuscript and genome sequence data was not officially accepted when we performed our genome analysis and prepared our manuscript. To prepare the revised version of our manuscript, we performed a genome alignment analysis using our assembly and that reported by Li *et al.* (2023). The results of this comparative are now included in Supplementary Fig. S5 as well as in the Result section of this revised version of our manuscript: “A genome sequence of another *S. hispanica* Mexican cultivar (white seeds) was recently published³⁹. This 4.6 Mb genomic fragment was also assigned to chr1 in this other *S. hispanica* genome, corroborating the result of our analyses. Genome alignment using this other Mexican cultivar also showed high collinearity between assemblies. Nevertheless, the results of this analysis indicate higher nucleotide identity between the assembly generated in this study with the Australian variety than the other Mexican variety published by Li *et al.*³⁹ (Supplementary Fig. S5).”

In general, both assemblies show significant collinearity, and the chia genome assembly reported by Li *et al.* also assigned this genomic fragment to chr1 (indicated as ‘gi|2’ in their assembly), confirming our results. We decided to mention such agreement among different

studies in the Discussion section: “Furthermore, a recently published chia genome, which is also at chromosome-level assembly, assigned this same genomic fragment to chr1 supporting our results.”

Comment Lines 141–147: *I presume this paragraph refers to gene expression in developing seeds, although this information is lacking. Presumably, this information could refer to developing seeds at various stages of post-fertilization seed development, mature seeds (which are mostly dormant for gene expression), or germinating seeds. Although Figure 2a provides some additional clarity, entirely in the image but not the caption, the reader should not have to try to decipher this information from Figure 2a. Instead, the authors need to clarify the developmental stage (or stages) of the seeds from which they obtained mRNAs in the narrative here, in the caption for Figure 2a, and in the Methods section. They also did not fully clarify to what extent their RNA-seq analysis is based on data from experiments they conducted and/or on data they obtained from previous publications, such as the research reported in their references 28–31. In referencing this previous RNA-seq research, the authors likewise did not clarify the developmental stage of the seeds analyzed in previous studies. I looked in the Methods section and could not find the information there. The fact that in lines 148–157 they refer to analysis of proteins and lipids from mature seeds further underscores the need for them to clarify the developmental stages of the seeds for RNA-seq analysis in lines 141–147.*

Response: We thank the reviewer for this important observation and comment. We modified Figure 2a to include a box with data labels and, in the figure legend, a brief description of the different tissue types and harvest day of each sample. In this revised version of our manuscript, we prepared an extended and detailed Methods section to clarify all the points kindly indicated by the reviewers, including all the RNA-seq data analyzed in the present study. The description we included in the Methods section is the following: “Publicly available RNA-seq data during *S. hispanica* life cycle including different tissues and growth stages²⁸ was retrieved from the EMBL-EBI ArrayExpress (experiment number E-MTAB-5515), and additional RNA-seq dataset of *S. hispanica* during seed development³¹ was retrieved from the NCBI SRA database (accession number PRJNA196477). Samples during the life cycle included: cotyledon and shoot at 3 days (3D); leaf primordia (LP) and shoot at 12D; 1st and 2nd leaves (P1P2), 3rd and 4th leaves (P3P4), 5th, 6th and 7th leaves (P5P6P7), and internode at 69D; top or bottom half raceme inflorescence at 158D; flowers at 159D; flowers at 164D; and dry seeds. Samples during seed development included: developing seeds at 3, 7, 14, 21, and 28 days after flowering. A detailed description of each tissue type and growth stages included in these transcriptomes is provided in **Supplementary Table S8.**”

The information between lines 141 to 147 refers specifically to dry chia seeds that are in a dormant state, but before reaching such state, they expressed and presumably accumulated several transcripts related to the acquisition of desiccation tolerance and preparation for the next stage during germination [as discussed by Dekkers *et al.* (2015); Verma *et al.* (2022)]. Indeed, the enrichment analysis of genes with high transcript levels in dry seeds (“**Supplementary Fig. S10**”) shows that they belong to categories related to embryo development ending and abiotic stress that are commonly reported related to seed desiccation tolerance acquisition.

Comment Lines 159–160: *Because these sentences begin a new section, the authors should clarify that the polyphenols to which they refer in this section are from mature seeds, not simply seeds (I presume this is the case, but I cannot be certain).*

Response: We thank the reviewer for this observation. The information in that statement needed to be completed. We modified the text in that section to be more specific: “Polyphenols were the fourth most abundant chemical group in mature seeds, with 15% of the chromatographic area.”

Comment Lines 179–182: *Here again the authors need to clarify the stage (or stages) of seed development. In line 180, they stated that “gene expression was examined across different tissues and stages.” However, in line 181, they state that “7 of these 113 genes had their maximum expression in seed” without clarifying which stage or stages of seed development they examined. They later, in lines 201–202 and 222–226, provided the sort of clarity regarding developmental stage that they need to provide here.*

Response: We analyzed differential gene expression to identify Chia genes involved in rosmarinic acid (RA) biosynthesis. Genes preliminary classified for the last two reactions of RA synthesis were filtered out by their expression in seeds compared to other tissues and developmental stages during *S. hispanica* life cycle. Therefore, we added the missing information in the paragraph of the main text describing Supplementary Table S8: “To better define candidates for the last two RA synthesis steps, gene expression was examined across different tissues and developmental stages (indicated as ‘life cycle’ samples in Supplementary Table S8)”. Additionally, we added a note in the figure legend indicating that sample descriptions and abbreviations are similar to those in Figure 2 (similar response to **Comment Lines 141–147**).

Comment Lines 205–207, 344, and 468–469: *The authors need to specify in statistical terms what they mean by “significant correlations.” Because this is a statistical term, they need to clearly state what they mean by “significant.” They need to describe the statistical test they used to derive this conclusion of significance and the probability threshold they chose for significance and why they chose this threshold. Figure 6 and its caption do not clarify the statistical nature of this correlation. In line 344, the authors used the relative phrase “more significant correlations” but failed to state the comparison. Do they mean “more significant correlations” relative to Arabidopsis or some other comparison, or do they simply mean “significant correlations”? The Methods section is also deficient in this respect. In lines 468–469, the authors simply state that “the output network was manually filtered to retain only significant correlations” without defining what a significant correlation is.*

Response: We thank the reviewer for this important comment. We agree with the reviewer that the term “significant” has statistical significance and that we incorrectly used it in describing our procedure to identify the potential regulators of mucilage production. In the revised version of our manuscript, we rephrased the text to avoid using the term “significant” incorrectly: “The RegEnrich package was used to construct gene regulatory networks with GRN (i.e., random forest algorithm) as the inference method and the list of predicted TFs as potential regulators. To select the important regulatory nodes, the output network object was ordered by edge weight and

filtered to retain the top 5% of the highest correlations. The resulting network was manually analyzed to identify all gene interactions involving mucilage-related genes, and finally, candidate regulators were identified by high node degree.

Comment Lines 287–293: *Here again the authors need to clarify the stage or stages of seed development to which they are referring in these sentences. I presume, but cannot be certain, that in lines 287–288 that are referring to developing seeds at some unidentified number of days post-fertilization, and in lines 289–293 to mature seeds, but I cannot be certain of these presumptions. They need to clarify the stages of the seeds throughout this paragraph.*

Response: Similar response to the **Comment Lines 141–147**. In the revised version of our manuscript, we included all the appropriate modifications to avoid misunderstanding in the analyses performed in this study.

Comment Lines 298–299: *Once again, the authors need to clarify that the seeds to which they are referring in this paragraph are mature seeds (if my presumption is correct).*

Response: We thank the reviewer for this observation. There needed to be more information in that statement to be clear. We added such information in this revised version of the manuscript: “Metabolome analysis of mature chia seeds determined that 15% of the total chromatographic area corresponds to polyphenolic compounds (Figure 2b).”

Comment Methods Section – *The authors need to add a paragraph describing their RNA-seq analysis in developing seeds and the preparation of plant material for this analysis, and how they analyzed RNA-seq data derived from other publications.*

Response: Similar response to the **Comment Lines 141–147**. We prepared an extended and detailed Methods section to clarify all the points kindly indicated by the reviewers, including all the RNA-seq data analyzed in the present study.

REVIEWER 2:

We thank this reviewer for the all the comments to improve our manuscript. The reviewer comments helped us to clarify the methodological approaches carried out in our study. In this revised version of our manuscript, we prepared and provided a complete description with all the details of the datasets and methodology performed in the present study. Additionally, the Discussion was edited to increase readability and clarity of the main findings of this study.

First Comment - *I did not see any information in the Materials and Methods section of the article on how batch effects between different transcriptome datasets from different sources were handled.*

Response: We thank the reviewer for this important observation and comment. In this revised version of our manuscript, we prepared an extended and detailed Methods section to clarify all

the points kindly indicated by the reviewers, including a description of all the RNA-seq data analyzed in the present study. A total of 66 publicly available RNA-seq libraries were used for the genome annotation process [Gupta *et al.* (2021); Sreedhar *et al.* (2015); Pelaez *et al.* (2019); Wimberley *et al.* (2020)]. For the approaches to study the molecular basis of specific chia traits such as rosmarinic acid pathway and mucilage-related genes, our analysis was limited to the datasets which include different tissues and stages during chia life cycle [Gupta *et al.* (2021)] or different points during seed development [Sreedhar *et al.* (2015)]. This modified Method section described in detailed the different RNA-seq datasets. We limited our gene expression analyses to RNA-seq libraries of the same dataset (we did not perform analysis combining the transcriptomes of the samples described 'life cycle' with the samples of 'seed development') to avoid the batch effects mentioned in the reviewer question. Then, in this modified version of the Methods, we indicate the samples used in each analysis ('life cycle' or 'seed development'):

"Publicly available RNA-seq data during *S. hispanica* life cycle including different tissues and growth stages²⁸ was retrieved from the EMBL-EBI ArrayExpress (experiment number E-MTAB-5515), and additional RNA-seq dataset of *S. hispanica* during seed development³¹ was retrieved from the NCBI SRA database (accession number PRJNA196477). Samples during life cycle included: samples from cotyledon and shoot at 3 days (3D); leaf primordia (LP) and shoot at 12D; 1st and 2nd leaves (P1P2), 3rd and 4th leaves (P3P4), 5th, 6th and 7th leaves (P5P6P7), and internode at 69D; top or bottom half raceme inflorescence at 158D; flowers at 159D; flowers at 164D; and dry seeds. Samples during seed development included: developing seeds at 3, 7, 14, 21 and 28 days after flowering. A detailed description of each tissue type and growth stages included in these transcriptomes is provided in **Supplementary Table S8**. RNA-seq raw data was quality trimmed (for adaptors, low-quality bases) using TrimGalore (v0.6.6). Gene expression analysis during chia life cycle was quantified using kallisto¹⁰¹ (v0.44), normalized (CPM). Average CPM value between replicates was transformed to Z score prior to principal component analysis (plotMDS limma package¹⁰²). (...) Expression levels of candidate RA genes during life cycle were quantified using kallisto¹⁰¹ (v0.44), and normalized using DESeq2 package¹⁰⁴. The mean expression across replicates was Z score normalized and used to analyze across different tissues and stages during life cycle. Previously reported genes for seed mucilage metabolism in *A. thaliana*^{30,57} were used to identify homolog genes in *S. hispanica* genome. Mucilage-related genes were identified using BLASTP with the set of *A. thaliana* genes as query. To avoid misidentification of mucilage-related genes, criteria to select homologs in *S. hispanica* included: bitscore ≥ 80 , alignment of the total length of the query sequence $\geq 80\%$, and identity between query and subject sequences $\geq 50\%$. Candidate genes were manually classified by their best hit according to the proposed function in *A. thaliana*. Expression levels of mucilage-related genes were quantified using kallisto¹⁰¹ (v0.44), and \log_{10} (TPM) values were Z score normalized to analyze expression during seed development."

Second Comment - It is not clear from the article how PCA analysis was performed on transcriptome data from different time points. Further information on the specific methods and procedures used would need to be provided to answer this question.

Response: We thank the reviewer for this comment. A description of how PCA analysis was performed is now included in the methods section of the revised version of our manuscript:

“Gene expression analysis during chia life cycle was quantified using kallisto¹⁰¹ (v0.44), normalized (CPM) and transformed to Z score prior to principal component analysis (plotMDS limma package¹⁰²).”

Third Comment - *The discussion section is too complicated and some parts can be merged into the results. Please simplify the discussion section.*

Response: We modified the discussion section for a better understanding of the aim, relevance and the approaches used in our study.

Fourth Comment – *Seed mucilage synthesis related genes were predicted in genome of chia (*Salvia hispanica*), but only identified the expanded gene family compared to *Arabidopsis thaliana* rather than its relative species.*

Response: We thank the reviewer for this interesting comment and suggestion. After the finding that mucilage-related genes were expanded in *Salvia hispanica* genome compared to *Arabidopsis*, we performed an exhaustive literature search to try to find other *Salvia* species which could produce a significant amount of mucilage in their seeds. Although *Salvia* genus comprises approximately 900 species, information about seed mucilage in other species is scant. We found a couple of literature reports which mention seed mucilage in just a few *Salvia* species, but to our knowledge, these species do not have a reported genome yet. We also tried but failed to get seeds to evaluate the presence of mucilage of one of the few *Salvia* species with a reported genome. In a further study, we will evaluate the expansion of mucilage synthesis genes in *Salvia* species or determine if this was a chia-specific evolutionary event.

Comment Line 92: *Please point out the concrete assembly that anchored into chromosomes.*

Response: The total assembly length was 351.98 Mb, and the assembly anchored into chromosomes was 345.52 Mb. The ‘Supplementary Table S5’ was slightly modified to emphasize the assembly that was anchored in the chromosomes, and we indicated such information in the Result section: “Most of the assembly was anchored in six pseudo-chromosomes (345.52 out of 351.98 Mb; Supplementary Table S5).”

Comment Line 103-106: *In addition to using Hi-C data to identify assembly errors, is it possible to supplement with ONT data to jointly determine assembly errors at this locus?*

Response: We thank the reviewer for this interesting comment. As we discussed in our manuscript, the biggest difference between our genome assembly and the assembly of the Australian variety was a fragment of 4.6 Mb. Our analyses indicated that this fragment belongs to chr1 and not to chr2 (as previously reported). Initially, we also considered the approach suggested by the reviewer to evaluate assembly errors at this locus. First, we analyzed if PacBio reads alignment to support this fragment position in the Australian genome, and there was no PacBio reads with a clear junction sequence to this fragment and the chr2 [as reported by Wang *et al.* (2022)]. A similar analysis using ONT reads in our assembly showed that some reads align

with the contigs in this controversial location. For this specific chromosome (chr1), the average ONT coverage is around 124x, and at this region, the coverage drops to around 57x. Our genome annotation process indicates high repetitiveness and low gene density in this Chr1 region; thus, probably this region is a large region with a high number of repeats that makes assembly difficult. We considered that few ONT reads were insufficient to assign this fragment's genomic location. Then, we validate the position of this genomic fragment using different Hi-C datasets, as explained in our manuscript. During our manuscript submission and reviewing process, another chia genome was published by Li *et al.* (2023). We performed a genome alignment analysis using our assembly and the recently reported by Li *et al.* (2023). Both assemblies show significant collinearity, and the chia genome assembly reported by Li *et al.* also assigned this genomic fragment to chr1 (indicated as 'gi|2' in their assembly), confirming our results. This is mentioned in the Discussion section of our revised manuscript: "Furthermore, a recently published chia genome, which is also at chromosome-level assembly, assigned this same genomic fragment to chr1 supporting our results.". We consider this a big difference between publicly available chia genome assemblies, and such information is missing in the recent article published by Li *et al.* (2023).

Comment Line 134-139: *I cannot understand the meaning of this sentence.*

Response: We thank the reviewer for the observations regarding this specific paragraph. This paragraph was meant to report a genome-wide evaluation of DNA methylation levels in all sequence contexts at the gene body boundaries (upstream and downstream from the coding sequence) and inside Gene and Transposable element bodies in the *S. hispanica* genome. In this revised version of our manuscript, the paragraph was edited to increase readability and clarity and also to reflect changes in **Supplementary Fig. S7** where now the representation of gene body and TE body methylation levels is reported as DNA methylation levels as a percentage instead of relative density fractions. The new paragraph in our revised text is the follows: "DNA methylation levels (%) for chia genes were evaluated and showed that low levels were observed at the boundaries of the gene bodies near translation start and stop codons in all sequence contexts, these levels increase in the gene body predominantly in the CG context and to a lesser extent in CHG and CHH methylation. In contrast, DNA methylation of repetitive DNA sequences such as transposable elements (TEs) showed higher DNA methylation levels in all three methylation contexts (Supplementary Fig. S7)."

Comment Line 141: *Insufficient evidence in gene expression analysis associated with seed development when identify the unique expression pattern. Why not implement the RNA-seq data based on more than one development stage because of these samples used in specific days after flowering (DAF).*

Response: We thank the reviewer for this meticulous observation. Related response to the first reviewer's comment regarding possible batch effects between different transcriptome datasets from different sources. We limited our gene expression analyses to RNA-seq libraries of the same dataset (we did not perform analysis combining the transcriptomes of the samples described 'life cycle' with the samples of 'seed development') to avoid batch effects using different datasets. The life cycle dataset [Gupta *et al.*, (2021)], which to our knowledge, is the

most comprehensive collection of chia transcriptomes, with 13 different samples, includes 3 replicates per sample sequenced using the Illumina platform HiSeq 2500, whereas the seed development dataset does not include replicates and libraries were sequenced using the Genome Analyzer Ix system. Due to the lack of replicates and the different sequencing platforms for the seed development transcriptomes, we decided to perform each of our gene expression analyses using only one of the datasets. In the methods section of this revised version of our manuscript, we indicated the samples used in each analysis.

Comment Line 144: *Why use CPM as the measure of gene expression? Perhaps FPKM and TPM would be more appropriate.*

Response: We evaluated both CPM and TPM. In our analyses, both normalization methods produce very similar results. Furthermore, we also evaluated other PCA analysis tools, and the seed sample always showed a distinct, very different expression pattern. We would like to share with the reviewer the following graph of a similar analysis using TPM with the R package PCAtools:

Comment Line 148-149: *What is the number of biological replicates for transcriptome, metabolome, and proteome? I did not find the corresponding quantity in the article.*

Response: We thank the reviewer for this comment. We modified 'Supplementary Table S5' to include the number of biological replicates per transcriptome sample. In this revised version of our manuscript, the number of replicates for metabolic, lipidomic, and proteomic analyses are properly indicated in the Method sections.

Comment Line 162-164: *Please provide relevant literature.*

Response: We thank the reviewer for this observation. In this revised version of our manuscript, we provide the reference of the most recent manuscript review, which summarizes the biosynthetic pathway of rosmarinic acid in plants and includes the results of recent molecular studies [Trócsányi *et al.* (2020)].

Comment Line 168-169: *Can LC-MS be used to validate substances not found in the metabolome?*

Response: We consider that the word 'identified' could have produced a misunderstanding, and we rephrased that specific line to: "Almost all RA precursors and intermediates (9 out of 10) were **detected** in our metabolomic analysis (Figure 3)." This conveys that while it can be challenging to detect all the metabolites within a specific metabolic pathway using untargeted metabolomic analysis, utilizing HPLC-MS proves to be an effective technique for the rosmarinic acid biosynthetic pathway. Through this method, we were able to detect and identify 9 out of 10 molecules of interest. Our chosen approach for metabolite extraction, using methanol, greatly

facilitated the recovery of these types of compounds. Subsequently, these compounds were efficiently separated via reverse-phase chromatography under low pH conditions. Furthermore, electrospray ionization in the MS source demonstrated higher efficiency when the molecule contained ionizable groups, such as hydroxyl, carbonyl, and phenolic groups. Collectively, these factors contributed to the analysis's sensitivity and enabled the coverage of the majority of compounds within the rosmarinic acid pathway.

Comment Line 174-177: *Please provide relevant literature.*

Response: We thank the reviewer for this comment. There was missing information about the references in that statement. The references of previously reported rosmarinic acid synthase genes are included in this revised version of our manuscript.

Comment Line 196-197: *What method was used to test the significance of this expansion?*

Response: We thank the reviewer for this comment. The term 'expansion' is a relative term, and some parts of our manuscript were probably unclear. Unfortunately, seed mucilage metabolism has been poorly studied in plants other than *Arabidopsis*. Therefore, the lack of information about the genes participating in seed mucilage production in other species limited our study to a comparative analysis with the *Arabidopsis* plant model. Furthermore, this study could establish the basis for the identification and characterization of mucilage-related genes for other commercially important crops. To avoid misunderstandings and clarify that the observed mucilage-related gene expansion in chia is relative to the number of genes reported in *Arabidopsis* we indicated such comparison in the revised version of the manuscript. For example, in the Abstract, we state now: "The chia genome exhibits a significant expansion in mucilage synthesis genes (compared to *Arabidopsis*), and gene network analysis revealed potential regulators controlling seed mucilage production.". We also clarify this point in the last paragraph of the Introduction: "The high-quality genome annotation revealed an expansion in the copy number of genes involved in mucilage biosynthesis compared to the genes previously reported in *Arabidopsis thaliana*."

Comment Line 200: *The expression of mucilage-related genes is important for the investigation of seed development in chia, but there were confused in the concrete sampling time (3DAF, 7DAF and 14DAF etc.)*

Response: We thank the reviewer for this important observation. In this revised version of our manuscript, the sampling times were properly indicated in the Results and Methods sections. Results: "Using available RNA-seq data, gene expression of mucilage-related genes was examined during seed development at 3, 7, 14, 21 and 28 days after flowering (DAF)." Methods: "Samples during seed development included: developing seeds at 3, 7, 14, 21 and 28 days after flowering."

REVIEWER 3:

We thank this reviewer for the comments and suggestions for the improvement of our manuscript. The Discussion section was modified to highlight improvements in our genome version compared to the previous one. Additionally, the *Salvia hispanica* genome methylation results were edited to increase readability and clarity as well as modifications in the **Supplementary Fig. S7**.

Comment 1: *As mentioned in introduction section, seed mucilage is a matrix composed of polysaccharides, surrounding the seed. if yes, mucilage is biosynthesized or accumulated in seed coat? it seems that transcriptomic, metabolic and proteomic data from the seed coat tissue. personally, I think that these data are critical and direct evidence to understand the molecular or genetic basis that why Chia seed produces mucilage.*

Response: We thank the reviewer for this suggestion. Generating transcriptomic, metabolic and proteomic data from the seed coat tissue is technically challenging. We think that there are several considerations to properly perform that kind of analysis. First, seed development in chia would have to be fully characterized (to identify at least the first stages of embryo development after pollination). Then, the mucilage secretory cells would need to be identified in *S. hispanica* seeds. Importantly, the mucilage secretory cell morphology must be characterized and determined if these cells have a similar distribution and behavior as reported in *Arabidopsis*. Indeed, our findings indicate some similarities in seed mucilage production between chia and *Arabidopsis*, at least at early seed development. Currently, we are working in the characterization of seed development in *S. hispanica*. However, there is still a lot of work to be done in this emerging crop.

Comment 2: *As authors claimed, the levels of genome DNA methylation (including CG, CHG and CHH) are so high in gene body regions compared to intergene regions and TE regions. This seems unusual.*

Response: We appreciate the concerns of the reviewer regarding this matter; in the text, we mention: "The density levels of DNA methylation were globally evaluated for chia genes; low methylation levels were observed at the gene bodies near translation start and stop codons predominantly in the CG methylation context with very low CHG and CHH methylation levels." , referring to the DNA methylation fractions inside and in the boundaries of gene bodies, that is in concordance with what has been previously reported for *Arabidopsis* and a wide number of angiosperm plants. We further mention that DNA methylation in transposable elements is significantly higher than in gene bodies, referring to **Supplementary Fig. S7**. "In contrast, DNA methylation of repetitive DNA sequences such as transposable elements (TEs) showed a significant increase in methylation density in all three methylation contexts. (TEs; Supplementary Fig. S7)."

We think the reviewer's concerns originated from the way relative methylation fractions were represented in **Supplementary Fig. S7** and because values in Y-axis suggested the opposite of our remarks in the text because we used different parameters during data normalization to obtain Methylation Ratios for both Gene body and TEs. Therefore, **Supplementary Fig. S7** was modified to show DNA methylation values as a percentage, resulting in a more intuitive

interpretation and comparison between DNA methylation states in Gene Bodies and TEs. Also, the main text was edited to reflect changes in **Supplementary Fig. S7** : “DNA methylation levels (%) for chia genes were evaluated and showed that low levels were observed at the boundaries of the gene bodies near translation start and stop codons in all sequence contexts, these levels increase in the gene body predominantly in the CG context and to a lesser extent in CHG and CHH methylation. In contrast, DNA methylation of repetitive DNA sequences such as transposable elements (TEs) showed higher DNA methylation levels in all three methylation contexts (Supplementary Fig. S7).”

Comment 3: *Is there any certain link between mucilage and protein identified from seed (shown in Table 1), These proteins are involved in mucilage biosynthesis, or they usually mix with mucilage?*

Response: We thank the reviewer for this interesting question. To our knowledge, this study will represent one of the most complete characterizations of the proteins associated with seed mucilage. We manually annotated most mucilage-associated proteins, and our results indicated a broad range of functions. The type of proteins identified in chia seed mucilage does not suggest they are directly involved in mucilage biosynthesis. Probably, many of the proteins associated with chia seed mucilage represent essential components for its structure and mechanical properties rather than its biosynthesis. The biological role of seed mucilage is out of the scope of the present study, but some chia mucilage-associated proteins, such as Dirigent proteins could have an important role for seed germination and plant establishment during this stage.

Comment 4: *In discussion section, author should highlight how improved genome data from previous one, only resolve the repeat regions? how about the 34,748 protein-coding genes, same to previous identification?*

Response: We thank the reviewer for this comment to significantly improve our manuscript. The importance of the molecular references generated in the present study compared to previous references is highlighted in this revised version of our manuscript. The main improvements included the relocation of a large chromosome fragment, resolution and identification of highly repetitive centromeric regions, and an improvement in genome annotation. For example, as suggested by the reviewer, we now indicated in the Discussion section the improvement in the gene models annotated in *S. hispanica* genome: “An improved annotation of the *S. hispanica* genome was obtained in the present study with the prediction of 34,748 protein-coding genes (compared to 31,069 genes reported in the Australian variety¹⁶) and an almost complete set of conserved embryophyte orthologs (97% complete BUSCOs compared to 93.2% of the PacBio assembly; Supplementary Fig. S6).”

REVIEWERS' COMMENTS:

Reviewer #1 (Remarks to the Author):

The authors have appropriately addressed my concerns in their rebuttal letter and have revised the manuscript to correct and clarify the issues that I raised in my review.

Reviewer #2 (Remarks to the Author):

I have no comments for the revised version.

Reviewer #3 (Remarks to the Author):

This revised version of this manuscript has been improved throughout. Overall, the revised manuscript is good and deserves to be published.